# Effects and functional mechanisms of digital fitness platforms on online fitness payment behavior under the perspective of "She-economy"

**Kaidi Zhang**[1], **Tao Yang**[1,2*], **Zhipeng Liu**[2], **Cuixia Yi**[3,4], **Yueyun Hou**[5], **Tianyi Wu**[1]

**1** School of Economy and Management, Xi'an Physical Education University, Xi'an, Shaanxi, China, **2** School of Economics and Management, Shanghai University of Sport, Shanghai, China, **3** School of Humanities and Social Science, Xi'an Jiaotong University, Xi'an, Shaanxi, China, **4** Physical Education School, Shaanxi Normal University, Xi'an, Shaanxi, China, **5** School of Leisure, Xi'an Physical Education University, Xi'an, Shaanxi, China

* 450380452@qq.com

## Abstract

Women have become an important force for economic development. Online fitness payment behavior is an important part of women's pursuit of "beautiful skin and healthy body", and is also an important embodiment of the integration of the digital industry and the fitness industry. Based on Stimuli-Organism-Response and Technology Acceptance Model theories, this study systematically explored the relationship and mechanism of action between digital fitness platforms and online fitness payment behavior through a survey of 259 women using Structural Equation Modeling and Fuzzy Set Qualitative Comparative Analysis. It was found that women's perceived use of digital fitness platforms was positively related to satisfaction, continued use intention, and online fitness payment behavior, except for the relationship between perceived usefulness and satisfaction, which was not significant. With the exception of satisfaction, which did not mediate between perceived enjoyment and online fitness payment behavior, satisfaction, and continued using intention mediated and chained mediation between the other independent variables and the dependent variable. The "comprehensive" digital fitness platforms were the most popular and favorite among women, followed by the "hedonic" and "pragmatic" platforms. On this basis, corresponding countermeasures and recommendations are proposed to promote the healthy, sustainable, and stable development of the digital fitness industry.

## 1. Introduction

"She-economy" is an important result of economic development and gender emancipation [1], it is an economic phenomenon formed in response to women's consumption characteristics [2], valuing the presentation of one's appearance [3] and social status [4], and has a profound impact on the transformation of the consumer market [5]. *Sporting Goods 2023: A Disorganized World Needs to Bounce Back* shows that women in China spend approximately 15 to 20 percent more than men on sports and fitness activities [6]. *Contemporary Women's*

**Data availability statement:** All relevant data are within the paper and its Supporting Information files. The minimal anonymised dataset necessary to study findings is also uploaded below: https://pan.baidu.com/s/1jJw4kYjpxAvYGAgFZqQmKA.

**Funding:** The author(s) received no specific funding for this work.

**Competing interests:** The authors have declared that no competing interests exist.

*Fitness Insights Report* shows that women personal trainer users make up five times the percentage of male users [7]. As it can be seen, female gym-goers have become the mainstay of the fitness population. Currently, major sports brands are entering the women's market. Sports industry segments have also been targeting the "She-economy". Women have become an important driver in promoting the quality and upgrading of sports consumption.

On 3 August 2021, the State General Administration of Sports issued *the 14th Five-Year Plan for Sports Development* to encourage the development of smart fitness and virtual sports, with the aim of promoting the integration of the fitness industry with the digital industry [8]. *Global Fitness Trends Survey 2024* reveals "mobile fitness apps" and "online personal trainers" among the top 20 popular fitness behaviors [9]. Digital fitness is becoming a popular form of physical activity [10] and is permeating the daily lives of people around the world [11]. Driven by national policy support, advances in digital science and technology, and the development of the fitness industry, a wide variety of digital fitness products and services are emerging, which not only break the spatial boundaries and redefine the freedom of fitness but also become an effective tool for individuals to participate in physical exercise and promote physical health [12]. Therefore, the study of women's digital fitness consumption is particularly important.

A digital fitness platform is a new type of fitness tool based on digital technology, which monitors the user's exercise data and physical condition in real-time by applying technologies such as mobile internet, big data, and artificial intelligence and provides the user with the latest health information [13], social interaction [14], dietary management [15] as well as fitness programs [16]. The main form of expression is the fitness app. Fitness apps such as Keep, Migu, and Xiaomi Sports are more popular in China, and fitness Apps such as My Fitness Pal and 30 Day Fitness are more popular in foreign countries [17]. Currently, most of the studies on digital fitness platforms have used the determinants of TAM(Technology Acceptance Model) as the independent variable to analyze its impact on continue using intention [18,19], community interaction [17,20], and experimental teaching [21,22], and a very small number of studies have used the determinants of TAM as the dependent variable to investigate the impact of the mindflow experience on fitness app design [23,24].

Therefore, based on the perspective of the "She-economy", this study takes SOR (Stimuli-Organism-Response) as the theoretical framework and adopts TAM core constructs and definitions in an attempt to clarify the influencing factors of online fitness payment behaviors to construct the influence mechanism between digital fitness platforms and online fitness payment behaviors, and on this basis, to put forward targeted policy recommendations for the digital transformation of the fitness industry. The purpose of this study is to (1) investigate whether the use of a digital fitness platform influences their online fitness payment behavior; (2) if so, what are the specific functional mechanisms of the effect; (3) what combinations of conditional variables explain online fitness payment behavior. The novelty of this study mainly includes the following three aspects: First, there is a lot of existing research on women's sports consumption, but only a small number of studies have focused on women's sports consumption under the influence of digital technology [25], and even fewer have studied the impact of women's online sports consumption behavior. This study can explore the mechanism of digital fitness platforms' influence on online fitness payment behavior from the perspective of "She-economy", thus enriching related research. Second, most of the previous studies explored the influencing relationship between variables in a quantitative way and proposed paths in a qualitative way. For example, Chung K et al. [26] explored how the sensory experience of VR viewers affects their intention to consume VR products and services using only confirmatory factor analysis, and based on the above results, a path discussion was conducted. This study used two quantitative analysis methods, SEM and fsQCA, to analyze

the influencing mechanisms and paths of digital fitness platforms and online fitness payment behaviors, which increased the scientific validity of the study. Third, the introduction of perceived enjoyment in this study further enriches the TAM and extends the application of SOR theory to women's consumption. This study consists of six core components: firstly, a conceptual model was designed based on the SOR theory, and hypotheses were formulated using the determinants of TAM and online fitness payment behavior as independent and dependent variables, respectively. Secondly, the methods of data collection, measurement, and analysis are presented. Thirdly, the SEM model was used to test the influence mechanism of the independent and dependent variables and the mediating effect of satisfaction and continued use intention. Fourthly, the group pathway of online fitness payment behavior was explored by group analysis with fsQCA. Fifth, the research results were discussed and interpreted. Sixth, based on the findings, conclusions were drawn, and relevant recommendations were made.

## 2. Context and theoretical background

### 2.1. The role of women's fitness and the She-economy in digital consumer behavior

Previous study reveals that, compared to men, women's fitness emphasizes more on the important functions of body sculpting, emotional expression, and psychological construction [27]. Women's fitness in China has been influenced by a combination of social culture [28] and national strategies [29], resulting in a number of issues with local characteristics that have attracted the attention of many scholars. The research topics are mainly divided into five categories: first, the strategies and paths of women's fitness actions in different classes [30,31]; second, the cognitive and perceptual logic construction of women's participation in fitness [32,33]; third, the function of fitness in promoting women's social status [34,35]; fourth, the impact of different fitness modes on women's physical health [36,37]; and fifth, women's motivation to participate in fitness and its transformation [38,39]. On the one hand, most of these studies are qualitative studies based on the discipline of sociology or quantitative studies based on the discipline of biology, and there is a lack of micro-level studies combining qualitative and quantitative studies with the disciplines of consumer behavior and psychology. On the other hand, the development of digital science and technology to empower various industries has become the inevitable path of industrial transformation and upgrading, but there has been little research on digital technology fuelling the women's fitness sector in China.

Although other countries have earlier conducted research on women's fitness based on consumer perspectives [40], most of the relevant studies are the results of traditional consumer behavior theories, exploring the influencing factors of women's fitness consumption behavior, mainly focusing on the analysis of consumer motivations and attitudes. Li et al. [41] explored the differences in consumer perceptions and attitudes toward virtual-physical integrated marketing based on the O2O model and examined the impact of consumer perceptions and attitudes on consumer intentions. Khan et al. [42] confirmed that women consumers have a higher interest in fitness. Of these, the quality of fitness center services positively affects women's satisfaction [43]. Meng et al. [44] credited fitness videos for helping women improve health literacy. While similar studies have refined the traditional consumer behavior research framework, more have yet to explore the mechanisms influencing women's online fitness behavior.

In 1999, Kathy Matsui and Goldman Sachs proposed a theory related to women's economy, namely "women's economics", which was first published in the British magazine The Economist and was mainly used to illustrate the economic benefits that women could obtain in the workplace [45]. After more than 20 years of development, the connotation and extension of

this theory have changed dramatically. Shi Q's idea has gained consensus among academics, who believe that because of women's esteem for consumption and the remarkable effect of driving the economy, the unique economic circle and economic phenomenon that has formed around women's financial management and consumption is called "She-economy" [46]. This study likewise draws on the above viewpoints for the definition of "She-economy". Currently, more businesses and industries have begun to define the consumer population from a women's perspective and develop products based on the needs of women. Most economic experts are of the view that women's economic self-reliance, surging consumer demand, and purchasing power are constituting a new point of economic growth.

## 2.2. SOR theory

Mehrabian, Russell first proposed the SOR theory, which consists of three dimensions, S (stimuli) - O (organism) - R (response), and the relationship between the three is that the external stimulus affects the internal state of the individual, which in turn drives the individual response [47]. SOR theory has been widely used in the field of consumer behavior [48]. The model suggests that an individual's buying behavior is induced by physiological and psychological factors within the consumer as well as stimuli from the external environment. Consumers are stimulated by various factors to produce psychological changes to generate purchase motives and ultimately implement purchase behavior. In recent years, SOR theory has been gradually applied to explore the impact of digital platform usage on users' consumption intentions and behaviors. For example, Hewei T et al. [49] explored the factors influencing users' continued purchasing intention for fashion products in social electronic commerce and confirmed that perceived value has a significant positive effect on continued purchasing intention. Li [50] constructs the influence mechanism of short video platform usage on users' purchase intention based on information quality theory and SOR theory.

In a similar vein, digital fitness platform use may also have some impact on online fitness payment behavior. This study adopts the SOR theory as a research framework, with the core concept that women's online fitness payment behavior is triggered by a stimulus that stems from women's intuitive experience of using a digital fitness platform, the intrinsic physiological and psychological factors of change that occur under the impetus of this stimulus, which in turn leads to purchasing intentions and behaviors.

## 2.3. Technology acceptance model

The technology acceptance model (TAM), developed and validated by Davis, is a tool for measuring user acceptance of computers, with perceived usefulness and perceived ease of use as its two specific variables [51]. Perceived usefulness reflects the extent to which the new technology improves an individual's job performance. Perceived ease of use reflects how easy it is for individuals to use the new technology. The core idea of the model is that there is an effect of an individual's perceived ease of use and perceived usefulness of an information system on an individual's attitudes and intentions.

TAM has yielded few results in the field of women's fitness [52], but as research continues, scholars have enriched and refined the TAM model by introducing perceived enjoyment in other domains. In the field of e-shopping, Lu and Su's research shows that when consumers make online purchases, the entertainment of the situation they are in will have a significant impact on their willingness to buy and thus make purchase decisions [53]. In the field of e-reading, Sun Q et al. [54] introduced perceived enjoyment and confirmed that it can positively influence readers' experiential feelings, which in turn promotes continuous reading. Also, Stokburger et al. [4] confirm that women are more sensitive to the uniqueness of goods,

hedonic, and status values. Research has shown that women are more concerned with sensory experiences in the process of consumption [55] and health experiences in the outcome of consumption [56]. For example, digital fitness platforms, such as the Keep app, produce sports derivatives and peripherals for events that are more attractive to women [57]. In addition, physiological studies have shown that fitness stimulates dopamine secretion to make people happy and relaxed, and digital fitness activities enrich the exercise experience and may also bring new Perceived enjoyment. Therefore, this study introduces perceived enjoyment into the TAM, where women consumers are prone to positive attitudes and willingness to use digital fitness platforms if they feel recreational, healthy, and fun, which in turn influences consumption decisions.

## 2.4. The conceptual framework

In summary, the above provides an in-depth discussion of the role of women's fitness and the "She-economy" in digital consumer behavior, as well as the theoretical basis of the "SOR theory" "TAM model", the theoretical basis, and the underlying context. Based on the above, a clear articulation of the relationships between key concepts will help to enhance the coherence and rigor of the theoretical model. The conceptual framework of this study is that digital fitness platforms act as a source of stimuli that generate perceivers after women's use, and these perceptions affect women's satisfaction and Continue using intention, which in turn affects online fitness payment behaviors. Therefore, the "S" framework of this study is an attribute of the digital fitness platform. A hybrid model is constructed by combining elements of the TAM and SOR models as the "O" framework. Females complete the "R" response by engaging in avoidance or proximity behaviors towards online fitness through the "S" and "O". A well-constructed diagram (Fig 1) allows for a clearer and more organized presentation of these elements, enabling a more intuitive understanding of their connections and logical relationships, and providing strong support for the following research to be carried out.

## 3. Hypothesis formation

### 3.1. TAM model and satisfaction, continue using intention

TAM is one of the most influential information systems acceptance models, which has been used to study the acceptance and behavioral patterns of different populations towards new technologies and products by analyzing individuals' perceived usefulness and perceived ease of use. A study by Li et al. [58] found significant gender differences in health information searches between men and women. Then, there may be gender differences in the use of digital fitness platforms as a vehicle for health information search. Therefore, it is of great theoretical

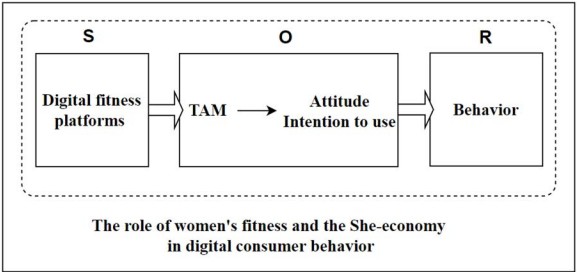

**Fig 1. The conceptual framework.**

and practical significance to study the impact of digital fitness platforms on women's online fitness payment behaviors in the context of "She-economy".

In the TAM model, perceived ease of use positively affects perceived usefulness and this relationship is stable [59]. The variables of TAM also affect individuals' satisfaction [60] and continue using intention [61]. Satisfaction is an expression of attitude that measures an individual's psychological response to a good, service, or experience, i.e., the difference between expectations and subjective feelings [62]. Continue using intention is a consumer's continued use of something rather than abandonment of its use. Zhan H [63] took college students as research subjects to explore their continue using intention online teaching platforms and found that perceived usefulness had a more pronounced effect on male students' continue using intention. Women college students' continue using intention the platform was more influenced by social factors. Tan H et al. [64] explored the effects of men's and women's student satisfaction and continue using intention with teachers online teaching, and the results showed that there were significant differences. Men students paid more attention to the external environment in their teachers' online teaching, while women students were more concerned with the learning outcomes and the quality of learning. Based on the TAM, Chen et al. [65] explored the behavioral characteristics and influencing factors of women's cross-border e-commerce consumption in China, and the study showed that the variables of the TAM had a significant impact on women's attitudes and willingness to cross-border e-commerce consumption. The TAM model of this study includes three determinants: perceived ease of use, perceived usefulness, and perceived enjoyment. Therefore, the following hypotheses are proposed:

H1a: Perceived ease of use positively and significantly affects satisfaction.

H1b: Perceived ease of use positively and significantly influences continue using intention.

H1c: Perceived ease of use positively and significantly affects perceived usefulness.

H2a: Perceived usefulness positively and significantly affects satisfaction.

H2b: Perceived usefulness positively and significantly influences continue using intention.

H3a: Perceived enjoyment positively and significantly affects satisfaction.

H3b: Perceived enjoyment positively and significantly influences continue using intention.

### 3.2. Satisfaction, continue using intention and online fitness payment behavior

In the context of the "She-economy", the diversified development of Internet products and services has drawn attention to women's emotional experience on the Internet. Satisfaction is regarded as a very important emotional variable that can positively influence women's intention and behavior [66]. Fitness behavior is women's purposeful and conscious use of free time under the interaction of intrinsic factors and the external environment to take the form and means of physical exercise, with the aim of physical and mental health or to achieve physical activities that are beneficial to people's health [67]. Online fitness payment behavior is the subjective activity of women who use the Internet to purchase fitness products and services in order to achieve fitness behavior [68]. Consistency of purchase and recommendations to others are important indicators of consumer behavior [69]. Low leisure time and low income are the two main reasons hindering women's fitness spending in China [70].

Women's satisfaction with a product or service is the main motivation for the continue using intention it [71] and an important factor influencing consumption behavior [72]. In the original TAM model and SOR theory, attitude and continue using intention have an impact on actual behavior. Lu Y et al. [73] prove the above point. Chen S et al. [74] explored the mechanism of single women's satisfaction on leisure tourism consumption behavior based on SOR theory. Based on the TAM theory, Miao S [75] found that women's perceived usefulness and perceived ease of use of gaming platforms have a significant effect on consumption intention. Mailizar et al. [76] verified that college students' attitudes toward using e-platforms significantly affect their intention to use them. Reiter et al. [77] similarly prove the above point. Therefore, the following hypotheses are proposed:

H4: Satisfaction positively and significantly influences online fitness payment behavior.

H5: Continue using intention positively and significantly influences online fitness payment behavior.

Drawing on previous research on "digital platforms", "She-economy", and "fitness consumption", this study examines the relationship between satisfaction, continuing using intention, and other concepts. While satisfaction and continue using intention have long been important variables for scholars studying digital platforms, little attention has been paid to their relationship with online fitness payment behaviors in the context of the "She-economy". In addition to this, the relationship between perceived enjoyment and the two variables needs to be examined. Therefore, a research model was constructed, as shown in Fig 2.

## 4. Methods

### 4.1. Measuring scales

This study constructed the questionnaire as the measure by utilizing established scales to maximize validity, which consisted of two parts. The first part is the respondent's basic personal information, which consists of four questions that include demographic characteristics such as age and education. The second part was the measurement of the variables of interest, which consisted of 18 questions. The research team followed the principles mentioned in the study of

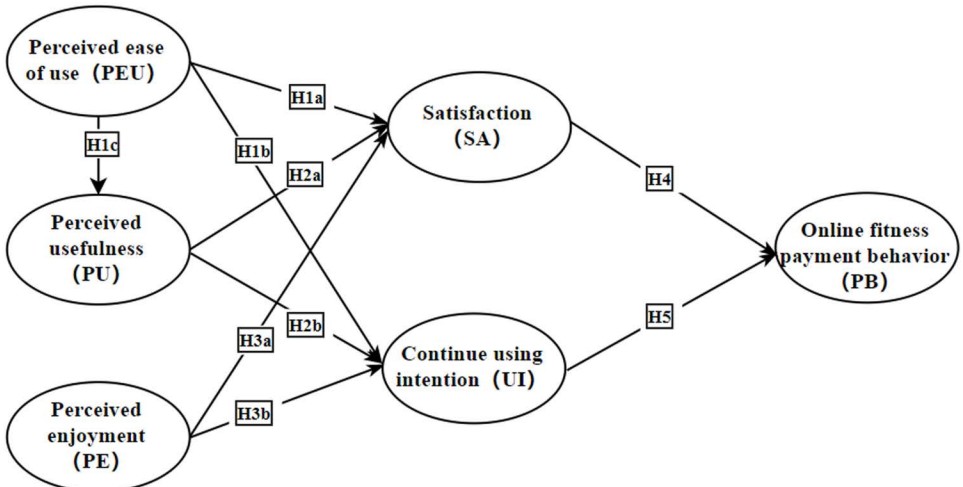

**Fig 2. Theoretical model of digital fitness platforms and online fitness payment behavior.**

Eisinga R et al. [78]. Poor quality items had to be removed from the limited pool of items, and the number of items in the scale could be two, as long as it was ensured that we were assuming that the available data were reasonable. The six items measuring perceived ease of use and perceived usefulness were taken from the Davis F D et al. [51] TAM scale. The four items for perceived enjoyment were taken from Sun Q et al. [54] Scale. The two items for satisfaction were borrowed from Wen X et al. [68] scale. The three items for continuing using intention were borrowed from Lin J S C et al. [79] scale, and the three questions on online fitness payment behavior are from the Jarvenpaa et al. [80] and Liu L et al. [81] scale. The Likert Scale was used, with 1 to 5 indicating "strongly disagree", "disagree", "generally", "somewhat agree", and "strongly agree".

## 4.2. Data collection

The questionnaire for this study was designed and completed by the author's team, and the data collection was done online and offline. Considering the availability and validity of the sample, the questionnaire was distributed from March to April 2024 using the snowballing method. Firstly, the author released questionnaires in Keep, Jitterbug, and WeChat's community interactions to conduct online surveys and randomly selected a group of women to conduct offline surveys in the Runners Without Borders Running Group, the Daqin Triathlon Club, and the Fanyino Pilates and Yoga Club; secondly, collected data through the family and friends of these respondents. A total of 323 questionnaires were recovered, 64 invalid questionnaires were eliminated, and finally, 259 valid questionnaires were obtained, with a recovery rate of 80.2% and the basic personal information statistics of the valid samples are shown in Table 1.

In terms of the age of the respondents, the lowest percentage was 0.39 percent for those younger than 18 years of age, and the highest was 84.17 percent for those aged 19-35 years. In terms of the educational level of the respondents, the lowest percentage was 6.18 percent

Table 1. Profile of respondents (N = 259).

| Variable | | Frequency | Percentages |
| --- | --- | --- | --- |
| Age | Less than 18 years | 1 | 0.39% |
| | 19-35 years | 218 | 84.17% |
| | 36-50 years | 27 | 10.42% |
| | Greater than 50 years | 13 | 5.02% |
| Educational background | Senior High School and below | 16 | 6.18% |
| | Bachelor's degree | 19 | 7.34% |
| | Bachelor's degree | 119 | 45.95% |
| | Master's degree or above | 105 | 40.54% |
| Frequency of use of digital fitness platforms | Never | 31 | 11.97% |
| | Sometimes | 143 | 55.21% |
| | Biweekly | 20 | 7.72% |
| | 2-3 times per week | 47 | 18.15% |
| | Almost every day | 18 | 6.95% |
| Duration of each use of digital fitness platforms | Less than 15 minutes | 59 | 22.78% |
| | 15-30 minutes | 95 | 36.68% |
| | 30-60 minutes | 83 | 32.05% |
| | 60-120 minutes | 20 | 7.72% |
| | More than 120 minutes | 2 | 0.77% |

for high school and below, the highest percentage was 45.95 percent for bachelor's degree, followed by 40.54 percent for postgraduate and above. In terms of frequency of use of digital fitness platforms, the lowest percentage of almost daily use was 6.95 percent, the highest percentage of occasional use was 55.21 percent, and based on the percentage of never use, it can be concluded that 88.03 percent of women have used digital fitness platforms; In terms of the duration of each use of digital fitness platforms, the lowest percentage was 0.77 percent for 120 minutes or more, the highest percentage was 36.68 percent for 15-30 minutes, followed by 32.05 percent for 30-60 minutes. The overall description of the women who use digital fitness platforms in this study shows that they are young, highly educated, low-frequency, and of appropriate duration.

### 4.3. Data analysis

A reliability test measures the reliability of the questionnaire. Cronbach α was used to test the reliability of the questionnaire. The result showed that Cronbach α for each variable was greater than 0.7, indicating that the questionnaire scale had good reliability.

The validity of the questionnaire was evaluated using a validity test employing KMO and Bartlett. The results showed that KMO value of 0.840 and Bartlett's p-value of 0. This indicates that the validity of the study data is high and suitable for factor analysis.

## 5. Results

### 5.1. SEM test

Exploratory factor analysis: A total of six common factors were rotated and extracted using the maximum variance method, with a cumulative variance explained rate of 76.739%, the common degree of each variable was above 0.6, the loading coefficients were above 0.7, and the rotated factor loading matrices were consistent with the expected factors. The results show that the validity between the variables and their corresponding question items meets the requirements of empirical analysis and can be tested in the next step.

Validation factor analysis: (1) Convergent validity. This study measures the convergent validity of the model through the CR value. The results show (Table 2) that the CR value of each variable is greater than 0.8, indicating good convergent validity. (2) Structural validity. The model's suitability to the data was judged through the comprehensive weighing and evaluation of multiple indicators [82]. The results showed that $\chi2/df = 1.202$, GFI = 0.943, NFI = 0.939, IFI = 0.989, TLI = 0.986, RMR = 0.049, RMSEA = 0.028, which indicated that the overall measurement model had a good fit. (3) Discriminant validity. In this study, measurements were made by AVE and Square root of AVE. As can be seen in Tables 2 and 3, the values of the variables are within the standard range, indicating good discriminant validity of the variables. In addition, Table 2 shows that the mean value of online fitness payment behavior (3.70) is the largest, indicating that women are willing to spend money on digital fitness platforms in the context of the "She-economy", while the mean value of perceived enjoyment (3.36) is the smallest, suggesting that women perceive digital fitness platforms to be less enjoyment.

SEM can identify, estimate and validate a variety of causal models, and has the advantage of completing the analysis of a complex model at one time and the test of the hypothesized model was the estimation of the path coefficients, which reflect the direction and degree of influence between potential variables. The conclusions are shown in Fig 3 and Table 4.

Firstly, the standardized path coefficients of perceived ease of use on satisfaction, perceived usefulness, and continue using intention are 0.263, 0.418, and 0.399, respectively, and all are significant, so hypotheses H1a, H1b, and H1c are validated, which suggests that an easy-to-use

**Table 2. Model validation results.**

| Variable | Question | Coefficients | Mean | CR | AVE |
|---|---|---|---|---|---|
| Perceived ease of use | PEU1 | 0.750 | 3.50 | 0.816 | 0.597 |
| | PEU2 | 0.817 | | | |
| | PEU3 | 0.841 | | | |
| Perceived usefulness | PU1 | 0.832 | 3.58 | 0.789 | 0.554 |
| | PU2 | 0.850 | | | |
| | PU3 | 0.744 | | | |
| Perceived enjoyment | PE1 | 0.828 | 3.36 | 0.901 | 0.696 |
| | PE2 | 0.859 | | | |
| | PE3 | 0.857 | | | |
| | PE4 | 0.816 | | | |
| Satisfaction | SA1 | 0.880 | 3.37 | 0.824 | 0.701 |
| | SA2 | 0.881 | | | |
| Continue using intention | UI1 | 0.842 | 3.41 | 0.849 | 0.653 |
| | UI2 | 0.784 | | | |
| | UI3 | 0.812 | | | |
| Online fitness payment behavior | PB1 | 0.855 | 3.70 | 0.853 | 0.660 |
| | PB2 | 0.861 | | | |
| | PB3 | 0.859 | | | |

**Table 3. Square root of AVE tests.**

| | PB | PE | UI | SA | PU | PEU |
|---|---|---|---|---|---|---|
| PB | 0.812 | | | | | |
| PE | 0.302 | 0.834 | | | | |
| UI | 0.265 | 0.428 | 0.808 | | | |
| SA | 0.207 | 0.335 | 0.300 | 0.837 | | |
| PU | 0.204 | 0.273 | 0.340 | 0.226 | 0.745 | |
| PEU | 0.201 | 0.339 | 0.477 | 0.334 | 0.330 | 0.773 |

digital fitness platform would positively affect women's favourability, perceived usefulness, and willingness to continue using it. The standardized path coefficient of perceived usefulness on willingness to continue using is 0.163 and significant, but the relationship with satisfaction is not significant, so hypothesis H2a is rejected and hypothesis H2b is accepted, which suggests that digital fitness platforms that are useful for women's fitness will positively affect women's willingness to continue using it but not their goodwill towards it. The standardized path coefficients of perceived interestingness on satisfaction and willingness to continue using are 0.260 and 0.290, which are significant. Therefore, hypotheses H3a and H3b are accepted, which suggests that an interesting digital fitness platform positively affects women's favorable perceptions of it and their willingness to continue using it. The above results confirm the findings of the study by Miao S [75] on the willingness of women to consume games.

The standardized path coefficient of satisfaction on online fitness payment behavior is 0.173 and significant, thus accepting hypothesis H4, which suggests that women's positive feelings towards digital fitness platforms positively influence their consumption behavior on the platform. The standardized path coefficient of willingness to continue to use on online fitness payment behavior is 0.271 and significant, hence hypothesis H5 is accepted, indicating

that women's willingness to continue to use digital fitness platforms positively influences their spending behavior on the platform. This is similar to the study of Reiter et al. [77].

### 5.2. Mediation effect test

When conducting the mediation effect test, in order to avoid the limitations of the conventional Sobel test, the Bootstrap method was used to test the mediation effect of satisfaction and continue using intention. Set up repeated sampling 5000 times to verify their mediating effects at 95% confidence intervals (95% CI). Since SEM verified that the direct effect relationship between perceived usefulness and satisfaction was not significant, there is no mediating effect. This may be because other factors have a more important influence between the two, such as perceived enjoyment. This study follows the steps of testing the new mediating effect of Bootstrap proposed by Wen Z et al. [83]. The testing process and results are shown in Table 5.

The results show that satisfaction and continue using intention have significant mediating effects between perceived ease of use and online fitness payment behavior, respectively. This suggests that easy-to-use digital fitness platforms stimulate women to feel good about it

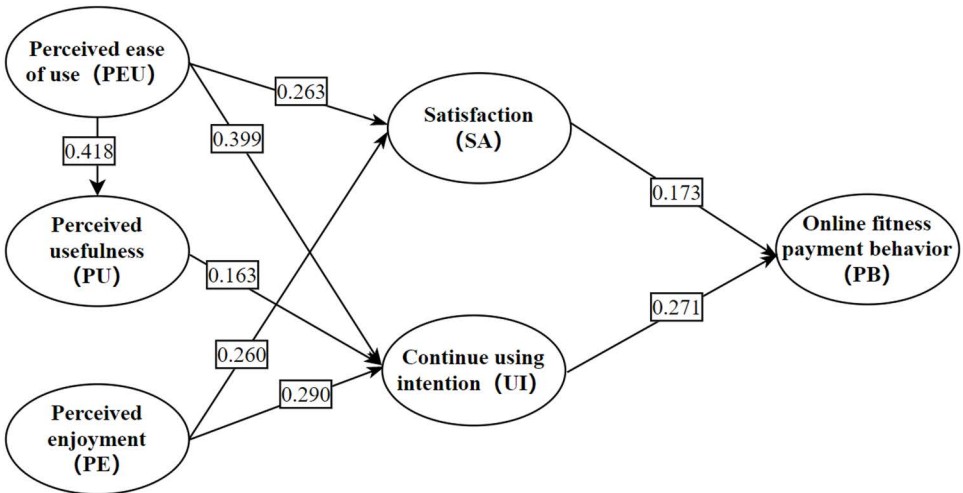

**Fig 3. Model of the impact of digital fitness platforms on online fitness payment behavior in the context of the "She-economy".**

**Table 4. Path coefficients and hypothesis testing results.**

| Path | | Coefficient | S.E. | C.R. | P Value | Conclusion |
|---|---|---|---|---|---|---|
| H1a | Perceived ease of use → Satisfaction | 0.263 | 0.109 | 2.942 | 0.003** | support |
| H1b | Perceived ease of use → Continue using intention | 0.399 | 0.067 | 4.868 | 0.000*** | support |
| H1c | Perceived ease of use → Perceived usefulness | 0.418 | 0.075 | 5.111 | 0.000*** | support |
| H2a | Perceived usefulness → Satisfaction | 0.106 | 0.108 | 1.315 | 0.188 | unsupport |
| H2b | Perceived usefulness → Continue using intention | 0.163 | 0.065 | 2.248 | 0.025* | support |
| H3a | Perceived enjoyment → Satisfaction | 0.260 | 0.098 | 3.424 | 0.001** | support |
| H3b | Perceived enjoyment → Continue using intention | 0.290 | 0.058 | 4.314 | 0.000*** | support |
| H4 | Satisfaction → Online fitness payment behavior | 0.173 | 0.061 | 2.273 | 0.023* | support |
| H5 | Continue using intention → Online fitness payment behavior | 0.271 | 0.090 | 3.584 | 0.000*** | support |

*p<0.05, **p<0.01, ***p<0.001.

**Table 5. Path coefficients for the mediated effects model.**

| Path | Effect | Coefficient | S.E. | 95% CI |
|---|---|---|---|---|
| Perceived ease of use→Satisfaction → Online fitness payment behavior | Aggregate effect | 0.179** | 0.054 | (0.072, 0.286) |
| | Indirect effect | 0.363*** | 0.064 | (0.238, 0.488) |
| | | 0.129* | 0.053 | (0.026, 0.233) |
| Perceived enjoyment → Satisfaction → Online fitness payment behavior | Aggregate effect | 0.303*** | 0.060 | (0.186, 0.420) |
| | Indirect effect | 0.409*** | 0.072 | (0.269, 0.550) |
| | | 0.098 | 0.052 | (−0.003, 0.199) |
| Perceived ease of use → Continue using intention → Online fitness payment behavior | Aggregate effect | 0.179** | 0.054 | (0.072, 0.286) |
| | Indirect effect | 0.326*** | 0.038 | (0.253, 0.400) |
| | | 0.285** | 0.089 | (0.111, 0.459) |
| Perceived usefulness → Continue using intention → Online fitness payment behavior | Aggregate effect | 0.209** | 0.063 | (0.086, 0.332) |
| | Indirect effect | 0.268*** | 0.046 | (0.178, 0.359) |
| | | 0.288** | 0.083 | (0.126, 0.450) |
| Perceived enjoyment → Continue using intention → Online fitness payment behavior | Aggregate effect | 0.303*** | 0.060 | (0.186, 0.420) |
| | Indirect effect | 0.330*** | 0.043 | (0.245, 0.416) |
| | | 0.216* | 0.085 | (0.050, 0.382) |
| Perceived ease of use → Satisfaction → Continue using intention → Online fitness payment behavior | Aggregate effect | 0.179** | 0.054 | (0.072, 0.286) |
| | Indirect effect | 0.363*** | 0.064 | (0.238, 0.488) |
| | | 0.290*** | 0.039 | (0.213, 0.367) |
| | | 0.100** | 0.036 | (0.029, 0.171) |
| | | 0.104** | 0.053 | (0.001, 0.207) |
| | | 0.255* | 0.090 | (0.079, 0.431) |
| Perceived enjoyment → Satisfaction → Continue using intention → Online fitness payment behavior | Aggregate effect | 0.303*** | 0.060 | (0.186, 0.420) |
| | Indirect effect | 0.409*** | 0.072 | (0.269, 0.550) |
| | | 0.285*** | 0.045 | (0.196, 0.374) |
| | | 0.111** | 0.037 | (0.038, 0.184) |
| | | 0.077 | 0.052 | (−0.025, 0.179) |
| | | 0.193* | 0.086 | (0.024, 0.361) |

*p<0.05, **p<0.01, ***p<0.001.

or willingness to use it consistently, which in turn leads to spending money on the platform. The mediating effects of satisfaction between perceived enjoyment and online fitness payment behavior are not significant. This suggests that while interesting digital fitness platforms can attract women's attention, it is more critical to have a direct effect on women's online fitness payment behavior rather than working through mediating effects. Continue using intention significantly mediates the relationship between perceived enjoyment and online fitness payment behavior. This suggests that when women develop a liking for the platform, there is a tendency to use it consistently, which in turn will promote payment behavior. Perhaps other aspects are having a more important impact on women's decision-making and behavior. continue using intention has a significant mediating effect between perceived usefulness and online fitness payment behavior. This suggests that a well-used digital fitness platform stimulates women to develop the idea of using it consistently and consequently spending money on the platform.

### 5.3. Fuzzy set qualitative comparative analysis test

Whereas SEM on the net influence relationships generated by the variables, fsQCA weakens their influence pathways and focuses on using set theory and Boolean algebra to identify the

causal influence of variable combinations with the outcome variable, which applies to the causal complexity problem designed for this study. Therefore, this study introduces fsQCA to explain the relationship between the effects of the variable further.

Variable selection and calibration. The results of SEM analysis revealed linear relationships between different variables, providing empirical evidence for exploring the mechanisms influencing digital fitness platforms and women's online fitness payment behaviors. Based on the literature basis and empirical findings, the determinants of TAM (perceived ease of use, perceived usefulness, and perceived enjoyment), satisfaction, and continue using intention were used as conditional variables, and online fitness payment behavior was used as an outcome variable. The dataset analyzed by fsQCA was calibrated to the data prior to specific analyzes [84], and based on the experience of previous studies [85], the present study used the quartile method to calibrate the variables involved as fully affiliated (75%), intersections (50%) and fully unaffiliated (25%). Fiss suggests subtracting a constant of 0.001 from the antecedent conditions with an affiliation of 1 or less to avoid data with the same degree of affiliation being removed from the analysis. The specific calibration values are shown in Table 6.

Analysis of necessary conditions. Before analyzing the sufficient conditions, it was necessary to test whether the conditioning variables were necessary for online fitness payment behavior (Table 7). Based on the experience of previous studies, the necessary conditions for a variable to be judged as an outcome variable are when its consistency and coverage are higher than 0.9 and 0.5 simultaneously [86]. The results showed that the consistency of the conditional variables for online fitness payment behavior ranged from 0.526-0.686, the coverage ranged from 0.642- 0.798, and the consistency of the individual conditional variables was all less than 0.9, which means that they do not constitute a necessary condition for high/low online fitness payment behavior, which indicates that a single conditional variable cannot fully explain the occurrence of women's online fitness payment behavior. Therefore, the conditional variables can be combined for further analysis to determine the effect of conditional groupings on individual online fitness payment behavior.

**Table 6. Variable calibration values.**

| Variable | PEU | PU | PE | SA | UI | PB |
|---|---|---|---|---|---|---|
| **Name after calibration** | fPEU | fPU | fPE | fSA | fUI | fPB |
| **Totally unaffiliated** | 1.750 | 2.000 | 2.000 | 1.450 | 1.333 | 1.970 |
| **Junction** | 3.500 | 4.000 | 3.333 | 3.500 | 3.667 | 3.667 |
| **Full affiliation** | 5.000 | 5.000 | 4.667 | 5.000 | 5.000 | 4.667 |

**Table 7. Analysis of requisites for online fitness payment behavior.**

| Variable | Consistency | Site coverage |
|---|---|---|
| **fPEU** | 0.676932 | 0.743396 |
| **~fPEU** | 0.534197 | 0.666997 |
| **fPU** | 0.596775 | 0.792194 |
| **~fPU** | 0.634508 | 0.662206 |
| **fPE** | 0.686183 | 0.769927 |
| **~fPE** | 0.526201 | 0.641505 |
| **fSA** | 0.667944 | 0.764195 |
| **~fSA** | 0.547413 | 0.653673 |
| **fUI** | 0.630146 | 0.797791 |
| **~fUI** | 0.654662 | 0.710332 |

Configuration analysis. The frequency threshold and consistency threshold need to be set when performing the group analysis, which were set to 3 and 0.85, respectively, with reference to the established studies [87], and this study chose intermediate solutions to analyze to make the results more complete and solvable [88]. As shown in Table 8, the overall coverage of the grouping that triggers women's online fitness payment behavior is 0.713144, and the overall consistency is 0.768606, indicating that the model has good interpretability.

Significantly, our study reveals a total of seven conditionally variable grouping paths that actively promote women's online fitness payment behavior. These paths, meticulously classified into three types of configurations based on the core and edge conditions, namely "hedonic", "comprehensive", and "pragmatic", hold crucial implications for understanding consumer behavior in the digital fitness platform landscape. The coverage of the "hedonic" pathway ranges from 0.362651 to 0.389480. The coverage of "integrated" pathways ranges from 0.435803 to 0.456552. The coverage of "pragmatic" pathways is 0.256658. This suggests that "comprehensive" digital fitness platforms are the most popular and favorite among women, followed by "hedonic" and "pragmatic".

In Y1, Y2, and Y3, satisfaction and perceived enjoyment are the core conditions, and perceived usefulness and continue using intention are the edge conditions and the three configurations are further named "hedonic" paths. Y1 and Y2 illustrate that when women perceive a digital fitness platform to be of little use and do not want to continue using it, they will nevertheless spend money on it if they have a strong liking for it or a higher level of interest in it. Y3 illustrates that when women perceive a digital fitness platform to be of little use, they will nevertheless spend money on it if they have a strong liking for it and a higher level of interest in it.

In Y4, Y5, and Y6, perceived enjoyment, satisfaction, perceived ease of use, and continue using intention are the core conditions, while the other conditions do not play a significant role, and they are further named as the "integrated" path. Y4, Y5, and Y6 indicate that when women find a digital fitness platform very easy to use, they will spend money on it if they have any 2 of the following 3 emotions at the same time, i.e., a strong desire to continue using it or a lot of good feelings or feeling that it is enjoyment.

In Y7, perceived usefulness is the core condition, and the other conditions are peripheral conditions, which are named "pragmatic" paths. Y7 says that when women find a digital fitness platform uninteresting and less favorable, they don't want to continue to use it, but if they find it useful, they will also engage in consumer behavior.

Robustness tests. In this paper, the method of increasing the consistency threshold was selected for re-analysis, increasing the consistency threshold from 0.85 to 0.9, and the frequency threshold remained at 3. The adjusted case coverage is slightly reduced, but the overall

**Table 8. Group analysis of factors influencing online fitness payment behavior.**

| | | Site coverage | Net coverage | Consistency |
|---|---|---|---|---|
| **Hedonic** | Y1:~ fUI*fSA*~fPU | 0.379435 | 0.0277539 | 0.850917 |
| | Y2:~ fUI*fPE*~fPU | 0.389480 | 0.0319832 | 0.849400 |
| | Y3: fSA*fPE*~fPU | 0.362651 | 0.00852436 | 0.831892 |
| **Synthesis** | Y4: fUI*fSA*fPEU | 0.435803 | 0.0234588 | 0.834810 |
| | Y5: fUI*fPE*fPEU | 0.456552 | 0.0361463 | 0.878226 |
| | Y6:fSA*fPE*fPEU | 0.436992 | 0.0239875 | 0.842742 |
| **Pragmatic** | Y7:~fUI*~f-SA*~fPE*fPU | 0.256658 | 0.0311240 | 0.847480 |
| Overall coverage: 0.713144 | | | | |
| Overall consistency: 0.768606 | | | | |

configuration and core conditions remain unchanged, so this paper's conclusions can be considered highly robust.

## 6. Discussion and conclusion

### 6.1. Discussion

**6.1.1. Influence relationships between variables.** The core factors of TAM have a significant positive effect on the satisfaction and continuing using intention of female users of digital fitness platforms, which in turn have a significant positive effect on online fitness payment behavior. That is, the higher the attributes of the core factors of TAM, the stronger the satisfaction and continuing using intention of the digital fitness platform for female users, which will further result in more consumption behaviors towards the digital fitness platform. This is consistent with Hasan's [89] study, which coincided with the fact that when women perceived online shopping to be less wise and less effective, women also had lower affective attitudes, including a preference for online shopping and feeling excited about it. From there, women also showed lower behavioral attitudes or behaviors. Therefore, as the digital economy and socio-cultural environment develop, the operability, usefulness, and enjoyment of digital fitness platforms will gradually increase, and women will increase their online fitness payment behavior. This result also validates Richard et al. [90] findings on women's online consumer behavior.

The perceived usefulness of female users of digital fitness platforms has little impact on their satisfaction. The underlying reasons for this are as follows: firstly, from a physiological point of view, women have lower extrinsic explosiveness and intrinsic testosterone hormones, which are important reasons for slow muscle growth [91]. Thus obtaining fewer positive incentives and lagging in the perceived usefulness for the software. Secondly, based on the compensation theory for an explanation, when women do not feel that the digital fitness platform can improve the quality of life, they will have a sense of frustration and look for new ways to make up for the loss; the enjoyment feeling appears just to make up for the lack of usefulness, thus forming compensation [92]. During the pre-survey period, it was found that more than 70% of women were interested in the medals of Keep software, "It's not important whether I can lose weight or not, but I'm determined to get the co-branded medals of Cherry Mariko, and I hope that I can come out with the co-branded models of my idols in the future", the above interview came from a master's degree student of the direction of the sports industry, B. It can be seen that there is a gap between the enjoyment and the usefulness of the platforms. There is a compensation mechanism between enjoyment and usefulness. Furthermore, women's consumption motives are easily influenced by factors such as fashion trends [93]. If they are involved in fitness to follow trends or keep up with their friends, the functionality and practicality of a digital fitness platform may not be their main concern, but more about whether the platform can provide a way to meet the current trends and thus satisfy their inner quest.

**6.1.2. Mediating role of satisfaction, continue using intention.** Continue using intention and satisfaction have mediating effects in the relationship between the core factors of TAM and online fitness payment behavior, respectively. That is, useful, enjoyment, and easy-to-use digital fitness platforms can increase women's excitement, generate satisfaction ratings or prolong women's stay on the platform, provide more traffic to the platform's push content, and thus increase the amount of time and money women spend on the platform. This is consistent with the study by Sewak [94] and Gueguen et al. [95] that when female consumers receive vivid information and have pleasant experiences while using an app, they rate the app's communication higher or spend more time on the app, which can lead

to increased consumption. This also follows the study by Le Mainardes, E.W. et al. [96] and Hong X et al. [97].

Perceived enjoyment can positively influence women's satisfaction, which in turn promotes women's online fitness payment behavior, but the mediating effect is not significant. Women may spend impulsively because of a strong excitement about the platform service, and are not overly concerned about satisfaction the digital fitness platform [98]. This explains why some women impulsively spend money on signing up for contests or events on platforms just to win fun prizes [99].

**6.1.3. The combined path of online fitness payment behavior.** The seven combinatorial variables of online fitness payment behavior were classified into three types of constructs, namely "hedonic", "integrative", and "pragmatic". The "hedonic" path suggests that women are more concerned with whether the services and products of digital fitness platforms can provide a pleasurable and enjoyable experience, as opposed to fitness outcomes. This further reflects the consumption characteristics of women [100], who tend to be easily attracted to services and products with novel and interesting designs when using digital fitness platforms [101]. They may impulsively consume because of an interesting challenge activity, a personalized training plan, or an attractive interface design on the platform [98]. However, this kind of impulsive consumption may also bring some problems, such as buying services or products that are not suitable for them because of their momentary interest, leading to waste [102]. This is consistent with what Cheng Q [103] found. The "comprehensive" path indicates that women focus on diversified and all-round experience, and they hope that the digital fitness platform can provide multi-element and multi-element courses and activities to meet different needs and interests. Liu C et al. [104] believe that with the improvement of consumption ability and upgrading of consumption concepts, consumers are more inclined to buy high-quality, high-value, and diversified goods. The "pragmatic" path has the lowest coverage rate, suggesting that a small number of women in the context of the "she-economy" are only concerned with the usefulness of digital fitness platforms but not the other technologies behind them. Women's physical attractiveness plays an important role in social behavior [105], and this pragmatic attitude may stem from women's direct need for physical attractiveness, with a greater focus on practicality and usefulness; this is consistent with the findings of the Hargreaves J's study [106].

## 6.2. Conclusion

This study proposes a hypothetical model of digital fitness platforms on women's online fitness payment behaviors based on a literature review and using SOR theory. By analyzing the survey data from 259 women, the following results were obtained: (1) Women's perceived ease of use of digital fitness platforms positively predicts perceived usefulness, satisfaction, and continue using intention, which in turn positively predicts online fitness payment behavior. Satisfaction and continue using intention have a mediating role in this process, respectively. (2) Women's perceived usefulness of digital fitness platforms does not predict satisfaction, but positively predicts continue using intention, and thus positively predicts online fitness payment behavior. continue using intention has a mediating role in this process. (3) Women's perceived enjoyment of digital fitness platforms can positively predict satisfaction and continue using intention, and thus positively predicts online fitness payment behavior. However, the mediating role of satisfaction and continue using intention is not significant. (4) There are seven conditioned variable grouping paths that can explain online fitness payment behavior, which can be classified into three types of "hedonic", "integrative" and "pragmatic" according to the core and edge conditions.

## 7. Implications and limitations

### 7.1. Implications

In terms of theoretical implications, this study first illustrates how women's use of digital fitness platforms influences their online fitness payment behavior, which will help us to understand women's online consumption behavior more comprehensively. Second, this is a new application of SOR theory and an extension of current research on TAM. Women will feel good about a digital fitness platform that is enjoyable, useful, and simple to operate, and they will want to continue to use it, thus spending money on that platform [107]. That is women's perceptions of what digital fitness platforms will do affect their satisfaction and continue using intention them, which in turn affects their online fitness payment behavior. However, no significant relationship was found between the usefulness of the platform and women's satisfaction, implying that women care less about the usefulness of the platform than about enjoyment and ease of use. Meanwhile, women are prone to paying for their interests and hobbies [108] and may spend impulsively to catch up with trends, leading to a less significant mediating role of satisfaction, and continue using intention between perceived enjoyment and online fitness payment behavior. Third, this study used fsQCA, a quantitative analysis method, to explore the pathways. The results revealed three configurations of group pathways. "Integrated" is the dominant pursuit of women. In other words, women seek platforms that provide "fitness benefits + emotional value". This was followed by "hedonic" and "pragmatic".

In a practical sense, the development of platforms needs to enhance their attributes. First of all, perceived ease of use is the most basic function of the platform, and they can make it easy for users to get started. Therefore, platform developers and administrators need to consolidate the ease of use of their own basic functions and provide users with efficient and convenient experiences in terms of UI interface, navigation structure, and accessibility. Secondly, perceived usefulness is the embodiment of the core functions of the platform, which can satisfy the initial intention of users to use it. On the one hand, artificial intelligence technologies such as big language models are actively applied. In the initial stage, the mining of user data is strengthened to accurately locate user needs and provide users with more personalized services; in the middle stage, the user data is trained on its own large model to provide users with more comprehensive body diagnosis and sports training planning; in the later stage, through the detailed analysis of the user's fitness characteristics and fitness habits, it is recommended for the user to recommend a more scientific, reasonable and diversified fitness In the later stage, through the detailed analysis of the user's fitness characteristics and fitness habits, we recommend more scientific, reasonable and diversified fitness programs and suggestions for users. On the other hand, user retention should be fully considered to improve the user stickiness of the product. In the initial stage, the initial accumulation of customer base can be achieved through simplifying the registration process, guided experience and free trial period of courses, equipment coupons, and incentives for reaching the target; in the middle stage, the customer base can be cultivated through personalized fitness plans, customized training courses, comment and praise interaction, and regular discounts; in the late stage, the long-term customer base can be enhanced through continuous improvement, optimization of functionality, return to rewards, community building, customer service, and other methods. Building, and customer service to enhance long-term retention.

The development of the platform also needs to emphasize exchanges and cooperation with other products, services, or resources. First, talent is the first resource for development. Therefore, cultivating talents who understand digital and sports is the first major event. Digital fitness platforms can realize the effective cycle of the three penetrating elements of education,

talent, and technology through policy support, school-enterprise cooperation, and other means, jointly cultivate talents that meet their own needs and provide guarantees for the continuous growth of the technical team and the platform itself through working practices such as platform design and operation. Secondly, perceived enjoyment is an important element that women are concerned about, which can attract female users and provide a more enjoyable fitness experience. On the one hand, through brand co-branding, supply chain integration, and other ways to expand the product field, continue to update the product content, and make use of the platform's social features to organize online and offline group activities to enhance its enjoyment. On the other hand, by keeping up with the trends and popular hotspots, we continue to provide product stories, provide users with more connotative products, and promote marketing. Third, cross-border circle-breaking cooperation should be deepened to improve the efficiency of platform usage. On the one hand, the platform can cooperate with professional fitness coaches, nutritionists, and rehabilitators to increase the usage rate of the platform, improve the exposure rate of knowledge providers, and innovate the platform's business model; on the other hand, the platform can also cooperate with sports apparel sales, cultural and creative product sales, and wearable social bloggers, to enrich the content of the platform, and to achieve the simultaneous development of such products through sports themes such as camping, marathon, and cycling.

## 7.2. Limitations

Despite the new findings of this study, there are still some limitations in the empirical research process. First, the relatively small sample size and limited scope of this study may make the participants not fully representative of the broader population in the context of the "She economy". Meanwhile, the cultural backgrounds of different countries and regions may also affect people's consumption behavior on digital fitness platforms. Therefore, future research could adopt more diverse sampling methods, expand the sample coverage, and consider cross-cultural research in order to increase the generalisability of the findings and better reveal the complex dynamics between the two. Second, this study uses self-reported data collection methods, which may lead to deviations between reported data and the actual situation, thus affecting the accuracy of the findings. In addition, the mediating effects of satisfaction and continue using intention between perceived enjoyment and Online fitness payment behavior were not significant and deserve to be explored further. Therefore, future research could adopt data collection methods such as observation, longitudinal research methods such as tracking surveys, and qualitative research methods such as semi-structured interviews to reduce data bias and explore consumers' long-term behavioral patterns on digital fitness platforms in depth and to further reveal the relationship between perceived enjoyment and other variables. Third, this study failed to adequately analyze the differences in online fitness payment behavior between groups in the context of the "She economy". Therefore, future research could divide the sample into "hedonic", "comprehensive", and "pragmatic" categories to clarify the group differences in online fitness payment behavior.

With the advancement of technology, especially the rapid development of artificial intelligence, its application in digital fitness scenarios is bound to change our fitness habits and ways. Consumers will be able, on the one hand, to conduct comprehensive body diagnostic consultations and customize personalized fitness programs through big language models in the fitness field, and on the other hand, they can use wearable devices such as VR and AR to learn fitness classes and carry out fitness exercises anytime, anywhere. However, the new technology may also create a digital divide, causing exercise constraints for groups that are relatively insensitive to electronic information devices. At

the same time, a large amount of Generative AI may also affect the quality of content in fitness platforms, and consumers' propensity to consume UGC and AIGC is still worth exploring.

## Supporting information

**S1 Table. All used data sets.** It contains the data used in the article.
(XLS)

## Acknowledgments

We are very grateful to the staff at Xi'an Physical Education University, Shanghai University of Sport, Xi'an Jiaotong University and Shaanxi Normal University.

## Author contributions

**Conceptualization:** Kaidi Zhang, Tao Yang.

**Data curation:** Yueyun Hou, Tianyi Wu.

**Formal analysis:** Yueyun Hou.

**Investigation:** Kaidi Zhang, Tianyi Wu.

**Methodology:** Kaidi Zhang, Zhipeng Liu.

**Supervision:** Tao Yang.

**Validation:** Cuixia Yi.

**Writing – original draft:** Kaidi Zhang.

**Writing – review & editing:** Kaidi Zhang, Zhipeng Liu.

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
