## [Decision Letter · Decision Letter 0]

27 Jun 2024

PONE-D-24-19773Effects and functional mechanisms of Digital Fitness Platforms on Online Fitness Payment Behavior under the Perspective of “She-Economy”PLOS ONE

Dear Dr. Yang,

Thank you for submitting your manuscript to PLOS ONE. After careful consideration, we feel that it has merit but does not fully meet PLOS ONE’s publication criteria as it currently stands. Therefore, we invite you to submit a revised version of the manuscript that addresses the points raised during the review process.

We look forward to receiving your revised manuscript.

Kind regards,

Kashif Ali, PH.D

Academic Editor

PLOS ONE

Journal Requirements:

2. You indicated that ethical approval was not necessary for your study. We understand that the framework for ethical oversight requirements for studies of this type may differ depending on the setting and we would appreciate some further clarification regarding your research. Could you please provide further details on why your study is exempt from the need for approval and confirmation from your institutional review board or research ethics committee (e.g., in the form of a letter or email correspondence) that ethics review was not necessary for this study? Please include a copy of the correspondence as an ""Other"" file.

3. In the online submission form, you indicated that "Data will be made available on request."

Additional Editor Comments:

Please addressed all the comments and suggestions given by the reviewers. 

Reviewers' comments:

Reviewer's Responses to Questions

**Comments to the Author**

1. Is the manuscript technically sound, and do the data support the conclusions?

Reviewer #1: Yes

Reviewer #2: Partly

Reviewer #3: Yes

2. Has the statistical analysis been performed appropriately and rigorously? 

Reviewer #1: Yes

Reviewer #2: I Don't Know

Reviewer #3: Yes

3. Have the authors made all data underlying the findings in their manuscript fully available?

Reviewer #1: Yes

Reviewer #2: No

Reviewer #3: Yes

4. Is the manuscript presented in an intelligible fashion and written in standard English?

Reviewer #1: Yes

Reviewer #2: No

Reviewer #3: Yes

5. Review Comments to the Author

Reviewer #1: The article explores the consumer market trends and behaviors in the context of women's fitness, emphasizing the significant role of women in driving sports consumption. It discusses the evolving perceptions of women's fitness, the influence of national policies, and the impact of digital fitness platforms on women's fitness behaviors.The development and design of digital fitness platforms require a combination of individual characteristics and product innovation to meet the needs of different users. Perceived ease of use is the most basic feature, making it easy for users to get started. Additionally, perceived enjoyment is crucial for engaging female users and providing a more enjoyable fitness experience. Developers should focus on consolidating the ease of use of basic functions and continuously expanding the product field through brand co-branding and supply chain integration. Enhancing user retention through personalized fitness plans and community-building activities is essential for the long-term success of digital fitness platforms.

Reviewer #2: 1) While it can be intuitive, it is not clear what is exactly meant by “She-economy”. A definition is missing.

2) What do TAM (line 67), SEM and fsQCA (line 84) stand for?

3) The citations in the text do not mention the year of publication. Sections are not numbered.

4) The SOR theory is not clearly explained, nor it is clear how it is used in the study.

5) Not clear what is meant here: “women’s psychological cognitive changes to online fitness occur after they feel internal and external stimuli using digital fitness platforms, which in turn affects women’s attitudes, intentions, and behaviors in fitness consumption”.

6) This is not clear: TAM affects individuals’ satisfaction and willingness to continue using it. TAM is a model and should not affect anything; it might be used to explain some relationships.

7) The 7 hypotheses feel redundant and repetitive. This should be simplified.

8) The definition of online payment behavior (line 224) is not clear. The literature that follows does not clarify it. Is payment behavior just the willingness to pay for a product/service?

9) The hypotheses from 10 to 18 are redundant, unclear and it is not discussed where they stem from.

10) Fig 1: everything seems to affect directly and indirectly anything else…

11) The data analysis should be better described and the results better explained and discussed. It is unclear whether the research questions presented in the introduction have been answered.

12) Overall, and looking at the title, it is not easy to understand what the contribution and the objectives of the paper are.

Reviewer #3: A literature search provides not only an opportunity to learn more about a given topic but provides insight on how the topic was studied by previous analysts. It helps to interpret ideas, detect shortcomings and recognize opportunities. In short, systematic and well-organized research may help in designing a novel research.

6. PLOS authors have the option to publish the peer review history of their article (what does this mean? ). If published, this will include your full peer review and any attached files.

**Do you want your identity to be public for this peer review?** For information about this choice, including consent withdrawal, please see our Privacy Policy .

Reviewer #1: **Yes: ** Assoc.prof.Dr.Nath amornpinyo

Reviewer #2: No

Reviewer #3: No

---

## [Author Response · Author response to Decision Letter 0]

18 Aug 2024

Journal Requirements:

Q1: Please ensure that your manuscript meets PLOS ONE's style requirements, including those for file naming.

Response: Thank you for this comment, and we have rearranged our manuscript to meet PLOS ONE's style requirements, including those for file naming.

Q2: You indicated that ethical approval was not necessary for your study. We understand that the framework for ethical oversight requirements for studies of this type may differ depending on the setting and we would appreciate some further clarification regarding your research.

Response: Thank you very much for your advice. I have submitted a certificate from my institution certifying that this study does not require ethical approval.

Q3: In the online submission form, you indicated that "Data will be made available on request."All PLOS journals now require all data underlying the findings described in their manuscript to be freely available to other researchers. either 1. In a public repository, 2. Within the manuscript itself, or 3. Uploaded as supplementary information.

Response: Thank you very much for your comment. I have uploaded the data as supplementary information.

Q4: PLOS requires an ORCID iD for the corresponding author in Editorial Manager on papers submitted after December 6th, 2016. Please ensure that you have an ORCID iD and that it is validated in Editorial Manager.

Response: Many thanks for this review. The corresponding author is registered with ORCID iD and has been verified in Editorial Manager.

Reviewer #1: 

Q1: The article explores the consumer market trends and behaviors in the context of women's fitness, emphasizing the significant role of women in driving sports consumption. It discusses the evolving perceptions of women's fitness, the influence of national policies, and the impact of digital fitness platforms on women's fitness behaviors.The development and design of digital fitness platforms require a combination of individual characteristics and product innovation to meet the needs of different users. Perceived ease of use is the most basic feature, making it easy for users to get started. Additionally, perceived enjoyment is crucial for engaging female users and providing a more enjoyable fitness experience. Developers should focus on consolidating the ease of use of basic functions and continuously expanding the product field through brand co-branding and supply chain integration. Enhancing user retention through personalized fitness plans and community-building activities is essential for the long-term success of digital fitness platforms.

Response: Thank you very much for your comment. Indeed, the paragraph is not precise enough. I have therefore made the following changes in line with your comments:

The development and design of digital fitness platforms require a combination of individual characteristics and product innovation to meet the needs of different users. Perceived ease of use is the most basic feature of the platform, and they make it easy for users to get started. Additionally, perceived enjoyment is crucial for engaging female users and providing a more enjoyable fitness experience. Therefore, digital fitness platforms need to consolidate the ease of use of their basic functions and provide users with an efficient and convenient experience in terms of UI interface, navigation structure, and accessibility. On the other hand, we also need to continuously expand the product field through brand co-branding and supply chain integration, continuously update the product content, and use the platform's social features to organize group activities online and offline to enhance its enjoyment. Enhancing user retention through personalized fitness plans and community-building activities is essential for the long-term success of digital fitness platforms.

Reviewer #2: 

Q1: While it can be intuitive, it is not clear what is exactly meant by “She-economy”. A definition is missing.

Response: Thank you for your comments and suggestions. We were indeed missing a definition of “her economy”. Therefore, we have added the following to this study.

She-economy

In 1999, Kathy Matsui and Goldman Sachs proposed a theory related to women's economy, namely ‘women's economics’, which was first published in the British magazine The Economist and was mainly used to illustrate the economic benefits that women could obtain in the workplace [42]. After more than 20 years of development, the connotation and extension of the ‘she economy’ have changed dramatically. Shi Qingqi's idea has gained consensus among academics, who believe that because of women's esteem for consumption and the remarkable effect of driving the economy, the unique economic circle and economic phenomenon that has formed around women's financial management and consumption is called “She-economy” [43]. This study likewise draws on the above viewpoints for the definition of “She-economy”. Currently, more businesses and industries have begun to define the consumer population from a women’s perspective and develop products based on the needs of women. Most economic experts are of the view that women's economic self-reliance, surging consumer demand, and purchasing power are constituting a new point of economic growth.

Q2: What do TAM (line 67), SEM and fsQCA (line 84) stand for?

Response: Thank you for your question. Indeed, the use of TAM (line 67) is not precise enough and SEM and fsQCA are not adequately described, which may lead to confusion. Therefore, we modified the description of TAM, SEM and fsQCA as follows:

Most of the studies on digital fitness platforms have used the determinants of TAM as the independent variable...

Structural equation modeling

Structural Equation Modelling (SEM) is an important statistical method for quantitative research in contemporary behavioural and social fields. It can identify, estimate and validate a variety of causal models, and has the advantage of completing the analysis of a complex model at one time, without having to split the model, and SEM has already become an obvious discipline in data analysis[54]. The common application for SEM is Amos.

In this study, SEM was applied to validate the hypothetical model diagram of the influential relationship between digital fitness platforms and online fitness payment behaviour.

Fuzzy set qualitative comparative analysis

Fuzzy set qualitative comparative analysis (fsQCA) is the incorporation of ideas from holism, group comparison theory, and set theory into the social science research context, examining the complex combinations of causes of social phenomena and the ways in which they influence them from a holistic and systemic perspective, aiming to uncover the combinations of antecedent conditions that lead to outcomes. In contrast to SEM, it distinguishes itself from both traditional qualitative research methods, which emphasise the interpretation, description, and prediction of individual cases or a small number of cases, and traditional quantitative research methods, which emphasise the statistical analysis of large samples of data, by exploring intermediate paths that circumvent the limitations of these two research methods [55]. There are three main types of qualitative comparative analysis methods, clear set qualitative comparative analysis (csQCA), multi-valued set qualitative comparative analysis (mvQCA), and fuzzy set qualitative comparative analysis (fsQCA). fsQCA avoids the limitations of csQCA and mvQCA, and does not need to take into account the degree of change of the research variables and the problem of partial affiliation. Therefore, fsQCA has received more attention in recent years in various research areas.

This study adopts the fsQCA method mainly for the following reasons: firstly, SEM-based empirical studies have verified the positive influence of digital fitness platforms on online fitness payment behaviour. However, this traditional quantitative research method is based on the assumption that the relationship between variables is linear, testing the net effect between variables, analysing the independent effects between various types of variables, and failing to analyse the combined effects between variables. The fsQCA method precisely studies the multiple causal relationships formed by different combinations of antecedent conditions from a group perspective, which provides a wider research paradigm on online fitness payment behaviour, and can, to a certain extent, complement SEM.

Q3: The citations in the text do not mention the year of publication. Sections are not numbered.

Response: Thank you for your suggestions regarding the citations in the article, but I apologise. Since PLOS uses the "Vancouver" style, which requires citing the reference number in square brackets, we did not add the year of publication in the in-text citations. Following your advice and the journal's requirements, we have listed the references at the end of the manuscript and numbered them in the order they appear in the text.

Q4: The SOR theory is not clearly explained, nor it is clear how it is used in the study.

Response: Thank you for your suggestions regarding the SOR theory. To make this study more rigorous and enhance the persuasiveness of the theoretical model, we have restructured the explanation of the SOR theory to make it clearer based on your suggestions and have included its application in the study.

Mehrabian, Russell first proposed the SOR theory, which consists of three dimensions, S (stimuli) - O (organism) - R (response), and the relationship between the three is that the stimulus affects the internal state of the individual, which in turn drives the individual response [44]. SOR theory has been widely used in the field of consumer behavior [45]. The model suggests that an individual's buying behaviour is induced by physiological and psychological factors within the consumer as well as stimuli from the external environment. Consumers are stimulated by various factors to produce psychological changes to generate purchase motives and ultimately implement purchase behaviour. In recent years, SOR theory has been gradually applied to explore the impact of digital platform usage on users' consumption intentions and behaviors. For example, Tian et al. explored the factors influencing users' continued purchasing intention for fashion products in social e-commerce and confirmed that perceived value has a significant positive effect on continued purchasing intention [46]. Li Xinying Constructs the Influence Mechanism of Short Video Platform Usage on Users' Purchase Intention Based on Information Quality Theory and SOR Theory [47].

In a similar vein, digital fitness platform use may also have some impact on online fitness payment behavior. This study adopts the SOR theory as a research framework, with the core concept that women's online fitness payment behaviour is triggered by a stimulus that stems from women's intuitive experience of using a digital fitness platform and the intrinsic physiological and psychological factors of change that occur under the impetus of this stimulus, which in turn leads to purchasing intentions and behaviours. Therefore, perceived ease of use, perceived usefulness, and perceived fun from the TAM model were used as the ‘S’ frame in this study, and the ‘O’ frame was introduced to indicate the psychological cognitive changes of women in terms of satisfaction and willingness to continue using. The ‘S’ and ‘O’ frames are used by women to generate avoidance or proximity behaviours toward online fitness, thus completing the ‘R’ response.

Q5: Not clear what is meant here: “women’s psychological cognitive changes to online fitness occur after they feel internal and external stimuli using digital fitness platforms, which in turn affects women’s attitudes, intentions, and behaviors in fitness consumption”.

Response: Thank you for your suggestions. We must apologize for the confused description in this section. Therefore, to be more concise and clear, we have rewritten this part according to your suggestions as follows：

Mehrabian, Russell first proposed the SOR theory, which consists of three dimensions, S (stimuli) - O (organism) - R (response), and the relationship between the three is that the stimulus affects the internal state of the individual, which in turn drives the individual response [44]. SOR theory has been widely used in the field of consumer behavior [45]. The model suggests that an individual's buying behaviour is induced by physiological and psychological factors within the consumer as well as stimuli from the external environment. Consumers are stimulated by various factors to produce psychological changes to generate purchase motives and ultimately implement purchase behaviour. In recent years, SOR theory has been gradually applied to explore the impact of digital platform usage on users' consumption intentions and behaviors. For example, Tian et al. explored the factors influencing users' continued purchasing intention for fashion products in social e-commerce and confirmed that perceived value has a significant positive effect on continued purchasing intention [46]. Li Xinying Constructs the Influence Mechanism of Short Video Platform Usage on Users' Purchase Intention Based on Information Quality Theory and SOR Theory [47].

Q6: This is not clear: TAM affects individuals’ satisfaction and willingness to continue using it. TAM is a model and should not affect anything; it might be used to explain some relationships.

Response: Thank you for your valuable suggestions and rigorous review. Indeed, the phrasing here was not accurate, so we have made the following revisions:

In the TAM, perceived ease of use positively affects perceived usefulness, and this relationship is stable [48,56]. These variables also affect individuals’ satisfaction [57] and continue using intention [58].

Q7: The 7 hypotheses feel redundant and repetitive. This should be simplified.

Response: Thank you for your suggestion. To explore the relationship and mechanisms between perception and behavior more thoroughly, we made comprehensive hypotheses based on relevant literature. However, some of the hypotheses were proved to be redundant and repetitive, and the description also needed to be clearer. Therefore, we have made the following revisions:

H1a: Perceived ease of use positively and significantly affects satisfaction.

H1b: Perceived ease of use positively and significantly influences continue using intention.

H1c: Perceived ease of use positively and significantly affects perceived usefulness.

H2a: Perceived usefulness positively and significantly affects satisfaction.

H2b: Perceived usefulness positively and significantly influences continue using intention.

H3a: Perceived enjoyment positively and significantly affects satisfaction.

H3b: Perceived enjoyment positively and significantly influences continue using intention.

Q8: The definition of online payment behavior (line 224) is not clear. The literature that follows does not clarify it. Is payment behavior just the willingness to pay for a product/service?

Response: Thank you for your advice. Indeed, we also consider that the definition of online fitness payment behavior in this study may be confused, and it should not merely represent payment intention. Therefore, based on your advice, we further clarified the concept of online fitness payment behavior as follows:

Fitness behaviour is people's purposeful and conscious use of free time under the interaction of intrinsic factors and the external environment, to take the form and means of physical exercise, with the aim of physical and mental health or to achieve physical activities that are beneficial to people's health [64]. Online fitness payment behaviour is the subjective activity of individuals who use the Internet to purchase fitness products and services in order to achieve fitness behaviour [65]. Consistency of purchase and recommendations to others are important indicators of consumer behaviour [66].

Q9: The hypotheses from 10 to 18 are redundant, unclear and it is not discussed where they stem from.

Response: Thank you very much for your valuable suggestion. Through detailed reading and discussion, we also consider that hy

---

## [Decision Letter · Decision Letter 1]

10 Sep 2024

PONE-D-24-19773R1Effects and functional mechanisms of Digital Fitness Platforms on Online Fitness Payment Behavior under the Perspective of “She-Economy”PLOS ONE

Dear Dr. Yang,

Thank you for submitting your manuscript to PLOS ONE. After careful consideration, we feel that it has merit but does not fully meet PLOS ONE’s publication criteria as it currently stands. Therefore, we invite you to submit a revised version of the manuscript that addresses the points raised during the review process.

**ACADEMIC EDITOR: **

Please address all the comments given by the reviewers. 

We look forward to receiving your revised manuscript.

Kind regards,

Kashif Ali, PH.D

Academic Editor

PLOS ONE

Reviewers' comments:

Reviewer's Responses to Questions

**Comments to the Author**

1. If the authors have adequately addressed your comments raised in a previous round of review and you feel that this manuscript is now acceptable for publication, you may indicate that here to bypass the “Comments to the Author” section, enter your conflict of interest statement in the “Confidential to Editor” section, and submit your "Accept" recommendation.

Reviewer #2: All comments have been addressed

Reviewer #4: All comments have been addressed

Reviewer #5: (No Response)

2. Is the manuscript technically sound, and do the data support the conclusions?

Reviewer #2: Yes

Reviewer #4: Partly

Reviewer #5: Partly

3. Has the statistical analysis been performed appropriately and rigorously? 

Reviewer #2: Yes

Reviewer #4: I Don't Know

Reviewer #5: Yes

4. Have the authors made all data underlying the findings in their manuscript fully available?

Reviewer #2: Yes

Reviewer #4: Yes

Reviewer #5: Yes

5. Is the manuscript presented in an intelligible fashion and written in standard English?

Reviewer #2: Yes

Reviewer #4: Yes

Reviewer #5: No

6. Review Comments to the Author

Reviewer #2: (No Response)

Reviewer #4: Thank you for the opportunity to review this manuscript. The title is intriguing, which set high expectations for the content. However, I found the article challenging to follow and understand in certain areas. While most of the previous reviewers' comments have been addressed, one recommendation—to number the sections—remains unimplemented.

Below are some additional suggestions that could further improve the manuscript:

Abbreviations: The manuscript uses abbreviations without first providing their full forms. For instance, the first usage appears in the Abstract. The previous reviewer likely meant that you should introduce the full term before using the abbreviation, rather than including extensive explanations of the concepts.

Literature Review:

Lines 84-85: The statement "there are few studies on women’s sports consumption" is debatable, as I have come across a number of studies in this area. It would be beneficial to clarify why the existing studies are deemed insufficient.

Lines 87-89: Consider providing specific examples of studies to support your statements.

Lines 125, 190: It would strengthen your argument to cite relevant studies here as well.

Writing Style:

Lines 167-168: Ensure that only the appropriate words are capitalized. Currently, it seems that words are unnecessarily capitalized, which might be incorrect.

Methods:

It is not clear how many questions were included in the questionnaire. This should be specified.

Lines 318-324: The source of the questionnaire sets should be clearly cited in square brackets, not just by mentioning the author. For instance, the citation "from Jarvenpaa and Others" is unclear—if "Others" is not a name, it should be lowercase, and typically, multiple authors are referred to as "Jarvenpaa et al." The source should also be included in the references list.

Results and Discussion:

The results appear somewhat intuitive. It would be useful to elaborate on the practical and theoretical implications of your findings. This will help readers understand the significance of your research.

The number of valid questionnaires analyzed is 259, not 323. I think this number should be mentioned in the Abstract.

The manuscript is somewhat wordy in places. Consider streamlining the text for clarity and conciseness.

The discussion section is underdeveloped, with results being compared to only three previous studies. A more thorough comparison with existing research is needed.

Conclusion:

The conclusion section is too lengthy and should not include subtitles. Additionally, the practical and theoretical implications of the study are not clearly outlined. This is a critical element that needs to be addressed.

In summary, while the manuscript presents an interesting topic, it requires significant revisions to improve clarity, conciseness, and scholarly rigor. I encourage the authors to consider these suggestions to enhance the quality and impact of their work.

I hope this review helps guide the necessary revisions for the manuscript.

Reviewer #5: General comment – First, I am pleased to have the opportunity to review this manuscript. The research presented aims to analyse the effects and functional mechanisms of Digital Fitness Platforms on Online Fitness Payment Behaviour under the perspective of the “She-Economy.” This topic is indeed compelling, especially considering the increasing influence of digital fitness platforms in the context of gender-targeted marketing, as noted by the authors. The study's relevance to current market trends makes it a valuable contribution to the literature.

However, I believe that the manuscript requires significant improvements before it can be considered for publication. While the authors have implemented changes from the previous review, several issues persist, particularly concerning the article's structure. The sections on theoretical background, hypotheses, discussion, and conclusion need substantial revision. The theoretical background should be more comprehensive and logically organised, providing a clearer foundation for the hypotheses. The discussion would benefit from a more thorough analysis of how the findings relate to existing literature, and the conclusion should better synthesise the study’s contributions and practical implications.

These structural improvements are essential for the manuscript to present a more cohesive and compelling narrative, effectively supporting the relevance and innovation of the research.

In the subsequent paragraphs, I will provide specific comments and concerns that arose during my review, which I hope will guide the authors in enhancing the quality of their work.

Comment 1 – The current title of the second section, "Research Basis", could be more informative. I suggest renaming this section to "Context and Theoretical Background." This revised title would better reflect the content, providing a clearer indication of what the reader can expect in this section.

Comment 2 – The subtitles "A Review of Research Related to Women’s Fitness" and "She-Economy" cover closely related themes and would benefit from being combined into a single subsection. I recommend a new title that highlights the integrated context of the research, such as "The Role of Women's Fitness and the She-Economy in Digital Consumer Behaviour." This would create a more cohesive narrative and emphasise the interconnectedness of these topics, allowing the reader to see the broader context of the study.

Comment 3 – I have observed that a previous reviewer requested clarification on the acronyms, "SEM" and "fsQCA", asking for their full forms. While this feedback highlights the need for clarity, it is not essential for the authors to provide detailed explanations of these methodologies within the second section of the manuscript. Instead, I recommend that the authors simply present the complete names of these methodological approaches the first time they are mentioned. For example, "SEM" as "Structural Equation Modelling" and "fsQCA" as "Fuzzy Set Qualitative Comparative Analysis."

These methodologies are well-established and widely recognised within the research community, and most readers familiar with the field will understand their significance without requiring extensive descriptions at this stage. Additionally, placing detailed methodological explanations in later sections where these methods are directly applied would enhance the manuscript’s structure and maintain the focus of the theoretical background. This approach will address the reviewer’s concern effectively while ensuring the manuscript remains clear and well-organised.

Comment 4 – The second section of the manuscript provides a thorough review of the relevant literature, discussing key concepts such as women’s fitness, the She-Economy, and the associated consumer behaviours. While this review is informative, the section would greatly benefit from the inclusion of a conceptual framework that synthesises the reviewed literature and sets the stage for the subsequent hypotheses.

A conceptual framework is essential as it visually and theoretically links the key variables discussed in the literature review, providing a clear and structured foundation for the research. By introducing a conceptual framework after the literature review, the authors can more effectively illustrate the relationships between the different variables and concepts, such as how the She-Economy influences online fitness payment behaviour through variables like satisfaction, perceived ease of use, and perceived usefulness.

This framework should serve as a bridge between the literature review and the hypothesis formation, making the logical flow of the manuscript more evident to the reader. It would also enhance the overall coherence of the research by providing a clear, visual representation of the theoretical model that underpins the study.

I strongly recommend that the authors develop and include a conceptual framework at the end of the second section. This will not only strengthen the theoretical grounding of the research but also provide a clear roadmap for the hypotheses and the subsequent empirical analysis.

Comment 5 – The current section titled "Theoretical Assumptions" appears overly simplified, particularly in its approach to forming hypotheses. The section would benefit significantly from incorporating more deductive reasoning derived from the theoretical framework presented earlier in the manuscript. Currently, the hypotheses seem to be stated without sufficient grounding in the theoretical background, which weakens the logical flow and the connection between the literature review and the hypotheses.

To enhance this section, I recommend that the authors first revise the title to better reflect its content and purpose. A more appropriate title could be "Hypothesis Formation," which clearly indicates that this section is dedicated to the development of hypotheses based on the theoretical framework. This title change will help align the reader’s expectations and clarify the section’s role within the manuscript.

Moreover, in forming the hypotheses, it is crucial to draw more explicitly from the theoretical background. The authors should aim to demonstrate how each hypothesis logically follows from the theories and concepts discussed in earlier sections. For example, when introducing the hypotheses related to perceived ease of use, perceived usefulness, and perceived enjoyment, the authors should first discuss these concepts within the context of the Technology Acceptance Model (TAM) and related literature, thereby providing a stronger justification for each hypothesis. Similarly, when discussing satisfaction, continue using intention, and online fitness payment behaviour, the authors should reference relevant theories and prior studies to build a more robust foundation for the hypotheses.

By integrating these elements into the hypothesis formation process, the section will not only become more comprehensive but will also provide a clearer rationale for the proposed research model. This approach will significantly improve the overall coherence and scholarly rigor of the manuscript.

Comment 6 – The discussion section of the manuscript, as it stands, is not sufficiently developed. It is important to emphasise that the discussion section plays a pivotal role in any research paper. This section is where the authors have the opportunity to showcase their critical thinking, derive solutions from their findings, and demonstrate a deeper understanding of the research problem. Currently, the discussion provided is largely descriptive and lacks the depth required to thoroughly engage with the implications of the findings.

To improve this section, the authors should focus on several key areas. First, the discussion should delve deeper into the broader implications of the findings. For instance, how do these results contribute to our understanding of women’s online fitness payment behaviour in the context of the She-Economy? What do these findings suggest for digital fitness platform developers, marketers, and policymakers? The discussion should not only reiterate the findings but also explore their significance in a broader context, comparing them with results from other studies and considering their practical applications.

Furthermore, the authors should engage more critically with the results, considering alternative explanations for the findings and discussing any limitations or unexpected outcomes. This critical analysis will demonstrate a more nuanced understanding of the research problem and provide a balanced view of the study’s contributions.

In summary, the discussion section should be expanded to include a more thorough analysis of the implications of the study, a comparison with other research, and a consideration of the broader impact of the findings. This will significantly enhance the manuscript’s overall quality and ensure that it provides valuable insights to the field.

Comment 7 – The conclusion section of the manuscript, as currently structured, does not adhere to the standard format typically expected in academic research papers. Several elements within this section would be more appropriately placed in the discussion or in a separate section dedicated to "Implications and Limitations."

Firstly, the content under "Focus on perceived enjoyment," "Enhancing user retention," and "Improving technology development" delves into detailed recommendations and implications that are more suited for the discussion section. The discussion is where these insights should be thoroughly explored, considering how the study’s findings contribute to existing knowledge and offering practical recommendations for industry professionals. By relocating this content to the discussion, the authors can provide a more comprehensive analysis of how these findings align with or diverge from other studies, thereby offering a deeper understanding of the research problem.

Secondly, the "Shortcoming" subsection, which addresses the limitations of the study and suggests areas for future research, should be separated into its own distinct section titled "Implications and Limitations." This section is crucial as it provides transparency about the study’s constraints and guides future research efforts. By clearly delineating these points, the manuscript will present a more organised and coherent structure, making it easier for readers to navigate the conclusions and implications of the study.

The conclusion itself should focus on summarising the key findings of the research, highlighting the main contributions to the field, and succinctly stating the broader implications without going into detailed recommendations or limitations, which should be reserved for the other sections mentioned above.

In summary, I recommend reorganising the current conclusion section by transferring the detailed implications to the discussion, creating a separate "Implications and Limitations" section, and refining the conclusion to be a concise summary of the study’s contributions.

7. PLOS authors have the option to publish the peer review history of their article (what does this mean? ). If published, this will include your full peer review and any attached files.

**Do you want your identity to be public for this peer review?** For information about this choice, including consent withdrawal, please see our Privacy Policy .

Reviewer #2: No

Reviewer #4: No

Reviewer #5: No

---

## [Author Response · Author response to Decision Letter 1]

6 Oct 2024

Reviewer #2: 

Response: Thank you for your support!

Reviewer #4: 

Q1: While most of the previous reviewers' comments have been addressed, one recommendation—to number the sections—remains unimplemented.

Response: Thank you very much for your valuable comments. Indeed, we have carefully revised and improved the comments of the previous reviewers. However, regarding the suggestion of numbering the chapters, it is true that we have not yet implemented it, for which we deeply apologise. We have now numbered the sections to enhance the clarity and conciseness of the article.

Q2: Abbreviations:

The manuscript uses abbreviations without first providing their full forms. For instance, the first usage appears in the Abstract. The previous reviewer likely meant that you should introduce the full term before using the abbreviation, rather than including extensive explanations of the concepts.

Response: Thank you for your comments. I strongly agree with your comments, so I have rechecked this study to align the abbreviations in the manuscript into line with the requirements for their use and to remove unnecessary explanations.

Q3: Literature Review:

Lines 84-85: The statement "there are few studies on women’s sports consumption" is debatable, as I have come across a number of studies in this area. It would be beneficial to clarify why the existing studies are deemed insufficient.

Lines 87-89: Consider providing specific examples of studies to support your statements.

Lines 125, 190: It would strengthen your argument to cite relevant studies here as well.

Response: Thank you very much for your advice, and I fully agree with your comments.. Indeed, my presentation was not precise enough. Therefore, I have reorganised this section in lines 84-85. In lines 87-89, I have added specific research examples to support my statement. Additionally, relevant studies have been cited in lines 125 and 190 to further strengthen the discussion. The following changes have been made:

Lines 84-85: First, there is a lot of existing research on women's sports consumption, but only a small number of studies have focused on women's sports consumption under the influence of digital technology [28], and even fewer have studied the impact of women's online sports consumption behaviour.

Lines 87-89: For example, Chung K et al. [29] explored how the sensory experience of VR viewers affects their intention to consume VR products and services using only confirmatory factor analysis, and based on the above results, a path discussion was conducted.

Lines 125: Although other countries have earlier conducted research on women's fitness based on consumer perspectives [40].

Lines 190: TAM has yielded few results in the field of women's fitness [52].

Q4: Writing Style:

Lines 167-168: Ensure that only the appropriate words are capitalized. Currently, it seems that words are unnecessarily capitalized, which might be incorrect.

Response: Thank you for your comment and I apologise for the oversight. I have now fixed the case of this part of the word.

Q5: Methods:

It is not clear how many questions were included in the questionnaire. This should be specified.

Lines 318-324: The source of the questionnaire sets should be clearly cited in square brackets, not just by mentioning the author. For instance, the citation "from Jarvenpaa and Others" is unclear—if "Others" is not a name, it should be lowercase, and typically, multiple authors are referred to as "Jarvenpaa et al." The source should also be included in the references list.

Response: Thank you very much for your suggestion and I strongly agree with you that it is necessary to clarify the number of questions in the questionnaire and provide the source. Therefore, I have added the number of questions and cited relevant references to clarify the source of the questionnaire, as modified below:

The first part is the respondent's basic personal information, which consists of four questions that include demographic characteristics such as age and education. The second part was the measurement of the variables of interest, which consisted of 18 questions. The research team followed the principles mentioned in the study of Eisinga R et al. [78]. Poor quality items had to be removed from the limited pool of items, and the number of items in the scale could be two, as long as it was ensured that we were assuming that the available data were reasonable. The six items measuring perceived ease of use and perceived usefulness were taken from the Davis F D et al. [51] TAM scale. The four items for perceived enjoyment were taken from Sun Q et al. [54] Scale. The two items for satisfaction were borrowed from Wen X et al.[68] scale. The three items for continuing using intention were borrowed from Lin J S C et al. [79] scale, and the three questions on online fitness payment behaviour are from the Jarvenpaa et al. [80] and Liu L et al. [81] scale.

Q6: Results and Discussion:

The results appear somewhat intuitive. It would be useful to elaborate on the practical and theoretical implications of your findings. This will help readers understand the significance of your research.

The number of valid questionnaires analyzed is 259, not 323. I think this number should be mentioned in the Abstract.

The manuscript is somewhat wordy in places. Consider streamlining the text for clarity and conciseness.

The discussion section is underdeveloped, with results being compared to only three previous studies. A more thorough comparison with existing research is needed.

Response: Thank you very much for your comment. I couldn't agree with you more.

(1)The results are indeed somewhat intuitive and it does help to elaborate on the practical and theoretical implications of the findings. Therefore, I first provided a detailed explanation of the research findings, and then added a revelation section at the end of the manuscript, where the theoretical and practical implications of the study were elaborated. The revisions are as follows:

7.1 Implications

In terms of theoretical implications. Firstly, this study illustrates how women's use of digital fitness platforms influences their online fitness payment behaviour, which will help us to understand women's online consumption behaviour more comprehensively. Second, this is a new application of SOR theory and an extension of current research on TAM. Women will feel good about a digital fitness platform that is enjoyment, useful, and simple to operate, and will want to continue to use it, thus spending money on that platform [101]. That is, women's perceptions of what digital fitness platforms will do affect their satisfaction and continue using intention them, which in turn affects their online fitness payment behaviour. However, it may be due to reasons such as women's presentation of fitness results being poorer than men's [102], thus producing a non-significant result between usefulness and satisfaction. Meanwhile, women are prone to paying for their interests and hobbies [103] and may spend impulsively to catch up with trends, leading to a less significant mediating role of satisfaction, and continue using intention between perceived enjoyment and online fitness payment behaviour. Third, this study used fsQCA, a quantitative analysis method, to explore the pathways. The results revealed three configurations of group pathways. “Integrated” is the dominant pursuit of women. In other words, women seek platforms that provide “fitness benefits + emotional value”. This was followed by “hedonic” and “pragmatic”.

In a practical sense, the development of platforms needs to enhance their attributes. First of all, perceived ease of use is the most basic function of the platform, and they can make it easy for users to get started. Therefore, platform developers and administrators need to consolidate the ease of use of their own basic functions and provide users with efficient and convenient experiences in terms of UI interface, navigation structure and accessibility. Secondly, perceived usefulness is the embodiment of the core functions of the platform, which can satisfy the initial intention of users to use it. On the one hand, artificial intelligence technologies such as big language models are actively applied. In the initial stage, the mining of user data is strengthened to accurately locate user needs and provide users with more personalized services; in the middle stage, the user data is trained on its own large model to provide users with more comprehensive body diagnosis and sports training planning; in the later stage, through the detailed analysis of the user's fitness characteristics and fitness habits, it is recommended for the user to recommend a more scientific, reasonable and diversified fitness In the later stage, through the detailed analysis of the user's fitness characteristics and fitness habits, we recommend more scientific, reasonable and diversified fitness programmes and suggestions for users. On the other hand, user retention should be fully considered to improve the user stickiness of the product. In the initial stage, the initial accumulation of customer base can be achieved through simplifying the registration process, guided experience and free trial period of courses, equipment coupons, and incentives for reaching the target; in the middle stage, the customer base can be cultivated through personalized fitness plans, customized training courses, comment and praise interaction, and regular discounts; in the late stage, the long-term customer base can be enhanced through continuous improvement, optimization of functionality, return to rewards, community building, customer service, and other methods. Building, and customer service to enhance long-term retention.

The development of the platform also needs to emphasise exchanges and cooperation with other products, services, or resources. First, talent is the first resource for development. Therefore, cultivating talents who understand digital and sports is the first major event. Digital fitness platforms can realise the effective cycle of the three penetrating elements of education, talent and technology through policy support, school-enterprise cooperation and other means, jointly cultivate talents that meet their own needs and provide guarantees for the continuous growth of the technical team and the platform itself through working practices such as platform design and operation. Secondly, perceived enjoyment is an important element that women are concerned about, which can attract female users and provide a more enjoyable fitness experience. On the one hand, through brand co-branding, supply chain integration and other ways to expand the product field, continue to update the product content, and make use of the platform's social features to organise online and offline group activities to enhance its enjoyment. On the other hand, by keeping up with the trend, popular hotspots, and so on, we continue to provide product stories, provide users with more connotative products, and promote marketing. Third, cross-border circle-breaking cooperation should be deepened to improve the efficiency of platform usage. On the one hand, the platform can cooperate with professional fitness coaches, nutritionists, and rehabilitators to increase the usage rate of the platform, improve the exposure rate of knowledge providers, and innovate the platform's business model; on the other hand, the platform can also cooperate with sports apparel sales, cultural and creative product sales, and wearable social bloggers, to enrich the content of the platform, and to achieve the simultaneous development of such products through sports themes such as camping, marathon, and cycling.

(2)As you suggested, the reference to the number of analysed valid questionnaires as 259 in the abstract is necessary. Therefore, I have revised the number of analysed valid questionnaires to 259 in the abstract.

(3)The manuscript section does seem to be a bit lengthy. Therefore, I have reorganised this section by removing and revising the lengthy areas to make it clear and concise.

(4)The discussion section did seem inadequate. Therefore I have rewritten this section to ensure a more comprehensive comparison with existing research. The revisions are as follows:

6.1 Discussion

6.1.1 Influence relationships between variables

The core factors of TAM have a significant positive effect on the satisfaction, and continuing using intention of female users of digital fitness platforms, which in turn have a significant positive effect on online fitness payment behaviour. That is, the higher the attributes of the core factors of TAM, the stronger the satisfaction, and continuing using intention of the digital fitness platform for female users, which will further result in more consumption behaviours towards the digital fitness platform. This is consistent with Hasan’s [89] study coinciding with the fact that when women perceived online shopping to be less wise and less effective, women also had lower affective attitudes, including a preference for online shopping and feeling excited about it. From there, women also showed lower behavioural attitudes or behaviours. Therefore, as the digital economy and socio-cultural environment develops, the operability, usefulness, and enjoyment of digital fitness platforms will gradually increase and, women will increase their online fitness payment behaviour. This result also validates Richard et al. [90] findings on women's online consumer behaviour.

The perceived usefulness of only female users of digital fitness platforms does not have a significant impact on their satisfaction. The author believes that there are the following reasons: firstly, from the physiological point of view, girls’ external explosive power, as well as internal testosterone hormone, are lower than boys’, which leads to less efficient fitness and muscle building than boys', thus obtaining less positive incentives, and lagging in perceived usefulness for the software; secondly, based on the compensation theory for an explanation, when women do not feel that the digital fitness platform can improve the quality of life, they will have a sense of frustration, and look for new ways to make up for the loss, the enjoyment feeling appears just to make up for the lack of usefulness, thus forming compensation. During the author's pre-survey period, it was found that more than 70% of women were interested in the medals of Keep software, “It's not important whether I can lose weight or not, but I'm determined to get the co-branded medals of Cherry Mariko, and I hope that I can come out with the co-branded models of my idols in the future”, the above interview came from a master's degree student of the direction of the sports industry, B. It can be seen that there is a gap between the enjoyment and the usefulness of the platforms. There is a compensation mechanism between enjoyment and usefulness. Furthermore, women's consumption motives are easily influenced by factors such as fashion trends [91]. If they are involved in fitness to follow trends or keep up with their friends, the functionality and practicality of a digital fitness platform may not be their main concern, but more about whether the platform can provide a way to meet the current trends and thus satisfy their inner quest.

6.1.2 Mediating role of satisfaction, continue using intention

Continue using intention and satisfaction have mediating effects in the relationship between the core factors of TAM and online fitness payment behaviour, respectively. That is, useful, enjoyment, and easy-to-use digital fitness platforms can increase women's excitement, generate satisfaction ratings or prolong women's stay on the platform, provide more traffic to the platform's push content, and thus increase the amount of time and money women spend on the platform. This is consistent with the study by Sewak [92] and Gueguen et al. [93] that when female consumers receive vivid information and have pleasant experiences while using an app, they rate the app's communication higher or spend more time on the app, which can lead to increased consumption. This also follows the study by Le Mainardes, E.W. et al. [94] and Hong X et al. [95].

Perceived enjoyment c

---

## [Decision Letter · Decision Letter 2]

5 Nov 2024

PONE-D-24-19773R2Effects and functional mechanisms of digital fitness platforms on online fitness payment behavior under the perspective of “She-economy”PLOS ONE

Dear Dr. Yang,

Thank you for submitting your manuscript to PLOS ONE. After careful consideration, we feel that it has merit but does not fully meet PLOS ONE’s publication criteria as it currently stands. Therefore, we invite you to submit a revised version of the manuscript that addresses the points raised during the review process.

**ACADEMIC EDITOR: **Please address all the comments given by the reviewers.

We look forward to receiving your revised manuscript.

Kind regards,

Kashif Ali, PH.D

Academic Editor

PLOS ONE

Journal Requirements:

Reviewers' comments:

Reviewer's Responses to Questions

**Comments to the Author**

1. If the authors have adequately addressed your comments raised in a previous round of review and you feel that this manuscript is now acceptable for publication, you may indicate that here to bypass the “Comments to the Author” section, enter your conflict of interest statement in the “Confidential to Editor” section, and submit your "Accept" recommendation.

Reviewer #4: All comments have been addressed

Reviewer #5: (No Response)

2. Is the manuscript technically sound, and do the data support the conclusions?

Reviewer #4: Yes

Reviewer #5: Yes

3. Has the statistical analysis been performed appropriately and rigorously? 

Reviewer #4: I Don't Know

Reviewer #5: Yes

4. Have the authors made all data underlying the findings in their manuscript fully available?

Reviewer #4: Yes

Reviewer #5: Yes

5. Is the manuscript presented in an intelligible fashion and written in standard English?

Reviewer #4: Yes

Reviewer #5: Yes

6. Review Comments to the Author

Reviewer #4: (No Response)

Reviewer #5: General comment – I appreciate the opportunity to review this revised manuscript. The authors have made meaningful progress, addressing several key points raised during the previous review. The structural changes, including the reorganisation of sections and the inclusion of a conceptual framework, have enhanced the manuscript’s clarity and alignment with academic conventions. The discussion section now offers a more in-depth engagement with the study’s findings and their implications for research and practice, which improves the overall narrative of the manuscript.

However, two areas still require further refinement before the manuscript can be considered for publication. Specifically, the theoretical model and the limitations section remain underdeveloped. While the revised framework provides a stronger foundation for hypothesis formation, some aspects of the theoretical linkages could benefit from greater elaboration. A clearer articulation of the relationships between key constructs would help strengthen the coherence and rigour of the theoretical model. Additionally, the limitations section, though expanded, could be further enriched by providing more detailed reflections on the study's constraints and suggesting more specific avenues for future research.

Comment 1 – While I appreciate the efforts to include a conceptual framework that integrates the S-O-R paradigm and TAM, I remain unconvinced that the TAM model should serve as the stimulation phase in the S-O-R paradigm. The S-O-R framework traditionally relies on external stimuli (e.g., environmental cues) to influence internal states and subsequent behaviours, whereas TAM primarily focuses on users' perceptions and attitudes towards technology adoption. Thus, positioning TAM as the stimulation phase may not fully capture the environmental influences intended in the original S-O-R model.

Instead, I recommend exploring the possibility of a hybrid model that synthesises elements from both TAM and the S-O-R paradigm. Such an approach would allow TAM’s constructs—such as perceived usefulness and perceived ease of use—to be incorporated within the organism or response phases, rather than as stimuli. This synthesis could enhance the conceptual model by acknowledging both environmental factors (stimuli) and users' internal evaluations (as part of the organism or behavioural response).

By adopting a hybrid framework, the study would offer a more nuanced understanding of how digital fitness platforms engage consumers, balancing technological acceptance with broader contextual and emotional factors. This approach would also align better with the study’s focus on behavioural outcomes, providing a clearer theoretical foundation for the subsequent hypotheses. I encourage the authors to reflect on this possibility and refine the conceptual framework accordingly to strengthen the coherence of the theoretical model.

Comment 2 – The limitations section, as it stands, provides only a surface-level overview of the constraints faced in this study. To enhance the academic rigour and transparency of the manuscript, I recommend that this section be further developed with more meaningful insights. In particular, the limitations should reflect not only the methodological constraints but also theoretical and practical challenges encountered during the research process.

For instance, the limited sample diversity and potential sampling biases could be discussed more explicitly—were the participants representative of the broader population within the She-Economy context? Additionally, it would be valuable to reflect on the generalisability of the findings across different geographical or cultural contexts, as digital fitness behaviour may vary significantly across regions.

The manuscript could also elaborate on limitations related to data collection, such as self-reported data biases or limitations in tracking behavioural patterns over time. Moreover, since the mediating effect of perceived enjoyment was found to be insignificant, the authors might discuss alternative methodological approaches—such as longitudinal studies or qualitative methods—that could provide deeper insights into this phenomenon in future research.

Finally, I suggest that the section conclude with a clearer outline of actionable directions for future research. In addition to investigating group differences and the mediating effects mentioned, future studies could explore how evolving technology trends (e.g., AI-based fitness recommendations) influence consumer behaviour within digital fitness platforms. Strengthening this section will not only improve the coherence of the manuscript but also demonstrate a deeper reflection on the limitations and their implications for future research.

7. PLOS authors have the option to publish the peer review history of their article (what does this mean? ). If published, this will include your full peer review and any attached files.

**Do you want your identity to be public for this peer review?** For information about this choice, including consent withdrawal, please see our Privacy Policy .

Reviewer #4: No

Reviewer #5: No

---

## [Author Response · Author response to Decision Letter 2]

8 Nov 2024

Dear Editor and reviewers:

On behalf of all the contributing authors, I would like to express our gratitude for your helpful feedback on our article. Your comments have been invaluable in improving our work. According to your suggestions, we have revised the manuscript to make our results more convincing. We have highlighted all the changes made to the manuscript using tracked changes in red. We offer a detailed point-by-point response to your insightful questions and suggestions below.

Editor:

Q1: Please review your reference list to ensure that it is complete and correct.

Response: Thank you for your valuable suggestion. To ensure the reference list is complete and correct, we reviewed all references in the manuscript. Firstly, we checked all references to see whether the article was retracted. Secondly, we re-edited the reference format of the article by Endnote to make it consistent with the journal’s requirements. In summary, the order of references has not changed, and the changes in reference formatting we have marked in red in the manuscript.

Reviewer #5: 

Q1：While the revised framework provides a stronger foundation for hypothesis formation, some aspects of the theoretical linkages could benefit from greater elaboration. A clearer articulation of the relationships between key constructs would help strengthen the coherence and rigour of the theoretical model.

Comment 1 – While I appreciate the efforts to include a conceptual framework that integrates the S-O-R paradigm and TAM, I remain unconvinced that the TAM model should serve as the stimulation phase in the S-O-R paradigm. The S-O-R framework traditionally relies on external stimuli (e.g., environmental cues) to influence internal states and subsequent behaviours, whereas TAM primarily focuses on users' perceptions and attitudes towards technology adoption. Thus, positioning TAM as the stimulation phase may not fully capture the environmental influences intended in the original S-O-R model.

Response: Thank you very much for your advice on the conceptual framework, which has been very enlightening. With your very professional knowledge, you gave us a perfect revision plan, which enhanced the completeness, logic, and coherence of the manuscript, and improved the quality of the manuscript, so thank you again. As you have stated, the use of the TAM model as the stimulus phase of the S-O-R paradigm in this study is controversial. The relationship between key concepts also needs to be further elaborated more clearly. Therefore, firstly, following your suggestion, I explored a hybrid model combining elements of the TAM and SOR models, with the TAM as part of the ‘O’. This enhances the conceptual model by recognising both environmental factors and users' internal evaluations. Secondly, the relationships between the key concepts in the framework have been analysed and elaborated in more detail, thus enhancing the coherence and logic of the article. The revisions have been highlighted in red in the manuscript. The specific changes are as follows:

In summary, the above provides an in-depth discussion of the role of women's fitness and the “She-economy” in digital consumer behaviour, as well as the theoretical basis of the “SOR theory” “TAM model”, the theoretical basis and the underlying context. Based on the above, a clear articulation of the relationships between key concepts will help to enhance the coherence and rigour of the theoretical model. The conceptual framework of this study is that digital fitness platforms act as a source of stimuli that generate perceivers after women's use, and these perceptions affect women's satisfaction and Continue using intention, which in turn affects online fitness payment behaviours. Therefore, the “S” framework of this study is an attribute of the digital fitness platform. A hybrid model is constructed by combining elements of the TAM and SOR models as the “O” framework. Females complete the “R” response by engaging in avoidance or proximity behaviours towards online fitness through the “S” and “O”. A well-constructed diagram (Fig 1) allows for a clearer and more organised presentation of these elements, enabling a more intuitive understanding of their connections and logical relationships, and providing a strong support for the following research to be carried out.

Q2: Comment 2 – The limitations section, as it stands, provides only a surface-level overview of the constraints faced in this study. To enhance the academic rigour and transparency of the manuscript, I recommend that this section be further developed with more meaningful insights. In particular, the limitations should reflect not only the methodological constraints but also theoretical and practical challenges encountered during the research process.

Response: Thank you for your valuable comments on the limitations section. Indeed, this section does not mention the theoretical and practical challenges, nor does it explicitly suggest directions in which future research could be conducted. I strongly agree with you that a more comprehensive account of the constraints of this study would add to the rigour and transparency of the manuscript. Therefore, firstly, I reflect on the possible generation of bias in sampling, including the range, number, and region of sample sources, and explore directions that could be improved in the future. Secondly, I reflect on the process of data collection and the results of data analysis, and explore the application of other methods in the future. Thirdly, we reflect on the group of people who pay for online fitness in the context of the ‘she economy’ and explore ideas for future research. Finally, I suggest a feasible direction for future research; the rapid development of technology brings different groups of people may have different impacts, resulting in consumption behaviours that are still worth exploring. At the same time, the large amount of Generative AI may also affect the quality of content in fitness platforms, and consumers' propensity to consume UGC and AIGC is still worth exploring. The specific changes are as follows:

Despite the new findings of this study, there are still some limitations in the empirical research process. First, the relatively small sample size and limited scope of this study may make the participants not fully representative of the broader population in the context of the “She economy”. Meanwhile, the cultural backgrounds of different countries and regions may also affect people's consumption behaviour on digital fitness platforms. Therefore, future research could adopt more diverse sampling methods, expand the sample coverage, and consider cross-cultural research in order to increase the generalisability of the findings and better reveal the complex dynamics between the two. Second, this study uses self-reported data collection methods, which may lead to deviations between reported data and the actual situation, thus affecting the accuracy of the findings. In addition, the mediating effects of satisfaction and continue using intention between perceived enjoyment and Online fitness payment behavior were not significant and deserve to be explored further. Therefore, future research could adopt data collection methods such as observation, longitudinal research methods such as tracking surveys, and qualitative research methods such as semi-structured interviews to reduce data bias and explore consumers' long-term behavioural patterns on digital fitness platforms in depth, and to further reveal the relationship between perceived enjoyment and other variables. Third, this study failed to adequately analyse the differences in online fitness payment behaviours between groups in the context of the “She economy”. Therefore, future research could divide the sample into “hedonic”, “comprehensive”, and “pragmatic” categories to clarify the group differences in online fitness payment behaviour.

With the advancement of technology, especially the rapid development of artificial intelligence, its application in digital fitness scenarios is bound to change our fitness habits and ways. Consumers will be able, on the one hand, to conduct comprehensive body diagnostic consultations and customise personalised fitness programmes through big language models in the fitness field, and on the other hand, they can use wearable devices such as VR and AR to learn fitness classes and carry out fitness exercises anytime, anywhere. However, the new technology may also creat a digital divide, causing exercise constraints for groups that are relatively insensitive to electronic information devices. At the same time, a large amount of Generative AI may also affect the quality of content in fitness platforms, and consumers' propensity to consume UGC and AIGC is still worth exploring.

---

## [Decision Letter · Decision Letter 3]

4 Dec 2024

PONE-D-24-19773R3Effects and functional mechanisms of digital fitness platforms on online fitness payment behavior under the perspective of “She-economy”PLOS ONE

Dear Dr. Yang,

Thank you for submitting your manuscript to PLOS ONE. After careful consideration, we feel that it has merit but does not fully meet PLOS ONE’s publication criteria as it currently stands. Therefore, we invite you to submit a revised version of the manuscript that addresses the points raised during the review process. **Comments from the editorial office** : 

Upon a closer evaluation of the manuscript, we have identified certain issues that we feel need to be addressed before this proceeds to publication, in order to comply with PLOS ONE's publication criteria (https://journals.plos.org/plosone/s/criteria-for-publication):

1. Ln 45, "Traditional thought, “soft beauty” is no longer the inherent label of women [8]. The Awakening of “Body Sense” [9], and The pursuit of “self-gratification, self transcendence and self-reading” is the “self-redemption” of women in the new era [10]" : It is unclear what this sentence means, and this would need to be clarified

2. ln 115, "Women’s fitness is a subordinate concept to fitness, with “body sensations” of pain, lightness, sweat, difficulty, heat, and being seen [9], emphasizing the important functions of body sculpting, emotional expression, and psychological construction [10]." : It is unclear what this sentence means, and this would need to be clarified

3. ln 118, "In the context of China’s socialist culture, women’s fitness has been influenced by the group’s body-shaping aspirations, the indulgence of social visual experience, and the advancement of the national fitness strategy, resulting in a number of research findings" : It is unclear what this sentence means, and this would need to be clarified. This sentence also needs to be supported by an appropriate reference.

4. Ln 124, "women’s fitness reconstruction of the gender order" : It is unclear what this sentence means, and this would need to be clarified.

5. Ln 246 - 250: It would be helpful if the authors briefly describe the differences, rather than only stating there was a difference

6. ln 539- ln 548: This seems to be the author's opinions only, and are unsupported by primary literature and as such would require clarification or revised to be removed

7. ln 575-ln 577: This seems to be the author's opinions only, and are unsupported by primary literature and as such would require clarification or revised to be removed

8. ln 585-592: This seems to be the author's opinions only, and are unsupported by primary literature and as such would require clarification or revised to be removed

9. ln 606 : It is unclear what this sentence means, and this would need to be clarified.

10. ln 634-635, "However, it may be due to reasons such as women's presentation of fitness results being poorer than men's": The article that has been referred to support this statement is not an appropriate one to support this. It is also further unclear what this means or what is meant by fitness results.

In addition to the concerns above, this manuscript also needs to be thoroughly copyedited for its English language usage and grammar, before this submission can be accepted, in order to comply with our publication criteria 5 (https://journals.plos.org/plosone/s/criteria-for-publication#loc-5).

We look forward to receiving your revised manuscript.

Kind regards,

Annesha Sil, Ph.D.

Associate Editor

PLOS ONE

On behalf of:

Kashif Ali, PhD

Reviewers' comments:

Reviewer's Responses to Questions

**Comments to the Author**

1. If the authors have adequately addressed your comments raised in a previous round of review and you feel that this manuscript is now acceptable for publication, you may indicate that here to bypass the “Comments to the Author” section, enter your conflict of interest statement in the “Confidential to Editor” section, and submit your "Accept" recommendation.

Reviewer #5: All comments have been addressed

2. Is the manuscript technically sound, and do the data support the conclusions?

Reviewer #5: Yes

3. Has the statistical analysis been performed appropriately and rigorously? 

Reviewer #5: Yes

4. Have the authors made all data underlying the findings in their manuscript fully available?

Reviewer #5: Yes

5. Is the manuscript presented in an intelligible fashion and written in standard English?

Reviewer #5: Yes

6. Review Comments to the Author

Reviewer #5: The revised manuscript demonstrates significant improvements, successfully addressing all the issues raised in previous review rounds. The refinement of the conceptual framework, particularly the integration of the TAM model as part of the "O" phase in the S-O-R paradigm, enhances its theoretical coherence and rigour. This adjustment, accompanied by a clearer articulation of the relationships between key constructs and the inclusion of a well-constructed figure, strengthens the manuscript’s contribution to the literature.

Furthermore, the expanded limitations section thoughtfully reflects on methodological constraints, theoretical challenges, and practical implications. The authors have also proposed insightful future research directions, including the role of generative AI in digital fitness platforms and its implications for consumer behaviour, which align well with emerging trends in the field.

The authors' diligence in incorporating feedback has resulted in a comprehensive and robust study, making a valuable contribution to the understanding of digital fitness platforms and the "She-economy."

7. PLOS authors have the option to publish the peer review history of their article (what does this mean? ). If published, this will include your full peer review and any attached files.

**Do you want your identity to be public for this peer review?** For information about this choice, including consent withdrawal, please see our Privacy Policy .

Reviewer #5: No

---

## [Author Response · Author response to Decision Letter 3]

8 Dec 2024

Dear Editors:

On behalf of all the contributing authors, I would like to express our gratitude for your helpful feedback on our article. Your comments have been invaluable in improving our work. According to your suggestions, we have extensively copyedited the English language usage and grammar our manuscript. We have highlighted all the changes made to the manuscript using tracked changes in red. We offer a detailed point-by-point response to your insightful questions and suggestions below.

Editors:

Q1: Ln 45, "Traditional thought, “soft beauty” is no longer the inherent label of women [8]. The Awakening of “Body Sense” [9], and The pursuit of “self-gratification, self transcendence and self-reading” is the “self-redemption” of women in the new era [10]" : It is unclear what this sentence means, and this would need to be clarified.

Response: Thank you very much for your comment. Indeed, the sentence was not expressed accurately enough, and the meaning was vague. Therefore, changes must be made. After checking the context of this sentence, we believe it is necessary to illustrate the increasing trend of women participating in fitness activities, which has drawn the fitness market’s attention to female consumers. Therefore, by modifying this sentence to “As it can be seen, female gym-goers have become the mainstay of the fitness population”, the logic of the passage can be made tighter and the language more condensed. The sentence will be more logical and concise.

Q2: ln 115, "Women’s fitness is a subordinate concept to fitness, with “body sensations” of pain, lightness, sweat, difficulty, heat, and being seen [9], emphasizing the important functions of body sculpting, emotional expression, and psychological construction [10]." : It is unclear what this sentence means, and this would need to be clarified.

Response: Thank you for your comment; it is your input that makes our research more rigorous. Indeed, the meaning of this sentence is not clear enough. Therefore, I think it is appropriate to revise this sentence to “Previous study reveals that, compared to men, women's fitness emphasizes more on the important functions of body sculpting, emotional expression, and psychological construction [27]”. This modification does not change the semantic meaning and makes the article clearer.

Q3: ln 118, "In the context of China’s socialist culture, women’s fitness has been influenced by the group’s body-shaping aspirations, the indulgence of social visual experience, and the advancement of the national fitness strategy, resulting in a number of research findings" : It is unclear what this sentence means, and this would need to be clarified. This sentence also needs to be supported by an appropriate reference.

Response: Thank you for your comment. This sentence is indeed inaccurate and also needs to be supported by an appropriate reference. Therefore, I modify this sentence to read “Women's fitness in China has been influenced by a combination of social culture [28] and national strategies [29], resulting in a number of issues with local characteristics that have attracted the attention of many scholars”, and added 2 references. The advantage of this is that it does not change the semantic meaning and is straightforward to express, so that readers can understand our work more clearly.

Q4: Ln 124, "women’s fitness reconstruction of the gender order" : It is unclear what this sentence means, and this would need to be clarified.

Response: Thank you very much for your comments. This sentence is not clear enough. Therefore, we revise the sentence to read “the function of fitness in promoting women's social status”. This isn’t change the meaning of the sentence and make the article clearer.

Q5: Ln 246 - 250: It would be helpful if the authors briefly describe the differences, rather than only stating there was a difference.

Response: Thank you very much for your comments. Yes, a brief description of the differences would be more helpful for readers to understand our work. The modifications are as follows:

Zhan H [63] took college students as research subjects to explore their continue using intention online teaching platforms and found that perceived usefulness had a more pronounced effect on male students' continue using intention. Women college students' continue using intention the platform was more influenced by social factors. Tan H et al. [64] explored the effects of men’s and women’s student satisfaction and continue using intention with teachers online teaching, and the results showed that there were significant differences. Men students paid more attention to the external environment in their teachers' online teaching, while women students were more concerned with the learning outcomes and the quality of learning.

Q6: ln 539- ln 548: This seems to be the author's opinions only, and are unsupported by primary literature and as such would require clarification or revised to be removed.

Response: Thank you again for your comments. The passage does lack support from the literature. Therefore, I have added references. The modifications are as follows:

The perceived usefulness of only female users of digital fitness platforms has little impact on their satisfaction. The underlying reasons for this are as follows: firstly, from a physiological point of view, women have lower extrinsic explosiveness and intrinsic testosterone hormones, which are important reasons for slow muscle growth [91]. Thus obtaining fewer positive incentives and lagging in the perceived usefulness for the software. Secondly, based on the compensation theory for an explanation, when women do not feel that the digital fitness platform can improve the quality of life, they will have a sense of frustration and look for new ways to make up for the loss; the enjoyment feeling appears just to make up for the lack of usefulness, thus forming compensation [92].

Q7: ln 575-ln 577: This seems to be the author's opinions only, and are unsupported by primary literature and as such would require clarification or revised to be removed.

Response: Thank you very much for your comments. Indeed, the passage lacks support from the literature. Therefore, I have added references. The modifications are as follows:

Women may spend impulsively because of a strong excitement about the platform service, and are not overly concerned about satisfaction the digital fitness platform [98]. This explains why some women impulsively spend money on signing up for contests or events on platforms just to win fun prizes [99].

Q8: ln 585-592: This seems to be the author's opinions only, and are unsupported by primary literature and as such would require clarification or revised to be removed.

Response: Thank you very much for your comments. Indeed, the passage lacks support from the literature. Therefore, I have added references. The sentence “Meanwhile, focusing too much on the hedonic experience may affect the achievement of the actual fitness effect to a certain extent” lacks support from literature and seems to be promoted based on individuals’ life experiences. Therefore, we consider that it would be more rigorous to delete this sentence. The modifications are as follows:

This further reflects the consumption characteristics of women [100], who tend to be easily attracted to services and products with novel and interesting designs when using digital fitness platforms [101]. They may impulsively consume because of an interesting challenge activity, a personalized training plan, or an attractive interface design on the platform [98]. However, this kind of impulsive consumption may also bring some problems, such as buying services or products that are not suitable for them because of their momentary interest, leading to waste [102].

Q9: ln 606 : It is unclear what this sentence means, and this would need to be clarified.

Response: Thank you very much for your comment. The sentence was indeed not clearly expressed. Therefore, I have rearranged the sentence here. The modifications are as follows:

This is consistent with the findings of the Hargreaves J’s study [106].

Q10: ln 634-635, "However, it may be due to reasons such as women's presentation of fitness results being poorer than men's": The article that has been referred to support this statement is not an appropriate one to support this. It is also further unclear what this means or what is meant by fitness results.

Response: Thank you very much for your advice. Yes, the sentence needs to be expressed clearly enough and have a logical relationship before and after. The reference citation is also inappropriate. Therefore, I have removed this sentence as well as this reference. The following sentence has been modified, but the semantic meaning remains the same. The purpose of this is to make the preceding and following content more logically coherent. The modifications are as follows:

However, no significant relationship was found between the usefulness of the platform and women's satisfaction, implying that women care less about the usefulness of the platform than about enjoyment and ease of use.

Several references were added and deleted as a result of the manuscript. Thus the reference order of the revised manuscript has changed compared to the original manuscript. We have marked the changes in red in the manuscript with track changes and listed the comparison of the reference order of the two manuscripts in the table below.

Number

In the Original Manuscript In the Revised Manuscript

[1]-[7] [1]-[7]

[8]-[9] deleted

[11]-[29] [8]-[26]

[10] [27]

- [28],[29]

[30]-[90] [30]-[90]

- [91],[92]

[91]-[95] [93]-[97]

- [98],[99]

[96] [100]

- [101],[102]

[97]-[101] [103]-[107]

[102] deleted

[103] [108]

---

## [Decision Letter · Decision Letter 4]

30 Jan 2025

Effects and functional mechanisms of digital fitness platforms on online fitness payment behavior under the perspective of “She-economy”

PONE-D-24-19773R4

Dear Dr. Yang,

We’re pleased to inform you that your manuscript has been judged scientifically suitable for publication and will be formally accepted for publication once it meets all outstanding technical requirements.

Kind regards,

Kashif Ali, PH.D

Academic Editor

PLOS ONE

Additional Editor Comments (optional):

Reviewers' comments:

Reviewer's Responses to Questions

**Comments to the Author**

1. If the authors have adequately addressed your comments raised in a previous round of review and you feel that this manuscript is now acceptable for publication, you may indicate that here to bypass the “Comments to the Author” section, enter your conflict of interest statement in the “Confidential to Editor” section, and submit your "Accept" recommendation.

Reviewer #6: All comments have been addressed

2. Is the manuscript technically sound, and do the data support the conclusions?

Reviewer #6: Yes

3. Has the statistical analysis been performed appropriately and rigorously? 

Reviewer #6: Yes

4. Have the authors made all data underlying the findings in their manuscript fully available?

Reviewer #6: Yes

5. Is the manuscript presented in an intelligible fashion and written in standard English?

Reviewer #6: Yes

6. Review Comments to the Author

Reviewer #6: The Authors have adequately addressed all of the comments and the manuscript is acceptable for publication

7. PLOS authors have the option to publish the peer review history of their article (what does this mean? ). If published, this will include your full peer review and any attached files.

**Do you want your identity to be public for this peer review?** For information about this choice, including consent withdrawal, please see our Privacy Policy .

Reviewer #6: **Yes: ** Essiagnon John-Philippe Alavo

---

## [Editor Report · Acceptance letter]

PONE-D-24-19773R4

PLOS ONE

Dear Dr. Yang,

I'm pleased to inform you that your manuscript has been deemed suitable for publication in PLOS ONE. Congratulations! Your manuscript is now being handed over to our production team.

Kind regards,

on behalf of

Dr. Kashif Ali

Academic Editor

PLOS ONE